



# Clustering analysis of very large measurement and model datasets on high performance computing platforms

Colin J. Lee[1], Paul A. Makar[1], Joana Soares[2]

[1]Air Quality Research Division, Environment and Climate Change Canada, Toronto, M3H 5T4, Canada
[2]The Climate and Environment Research Institute NILU, Kjeller, 2007, Norway

*Correspondence to*: Colin J. Lee (colin.lee@ec.gc.ca)

**Abstract.** Spatiotemporal clustering of data is an important analytical technique with many applications in air quality, including source identification, monitoring network analysis and airshed partitioning. Hierarchical agglomerative clustering is one such algorithm, where sets of input data are grouped by a chosen similarity metric, without requiring any *a priori*

information about the final arrangement of clusters. Modern implementations of the algorithm have $O(n^2 \log{(n)})$ computational complexity and $O(n^2)$ memory usage, where n is the number of initial clusters. This dependence can strain the resources of even very large individual computers as the number of initial clusters increases into the tens or hundreds of thousands, for example, to cluster all the points in an air-quality model's simulation grid as part of airshed analysis (~$10^5$ to $10^6$ time series to be clustered). Using two parallelization techniques – the Message Passing Interface (MPI) and Open Multi-

Processing (OpenMP) – we have reduced the amount of wallclock time while increasing the memory available to a new hierarchical clustering program, by dividing up the program into blocks which are run on separate CPUs but communicate with each other to produce a single result. The new algorithm opens up new directions for large data analysis which had previously not been possible. Here we present a massively parallelized version of an agglomerative hierarchical clustering algorithm which is able to cluster an entire year of hourly regional air-quality model output (538x540 domain; 290,520

hourly concentration timeseries) in 12 hours 37 minutes of wallclock time, by spreading the computation across 8000 Intel® Xeon® Platinum 8830 CPU cores with a total of 2TB of RAM. We then show how the new algorithm allows a new form of air-quality analysis to be carried out starting from air-quality model output. We present maps of the different airsheds within the model domain, identifying equally unique regions for each chemical species. These regions can be used as an aid in determining the placement of surface air-quality monitors which gives the most representative sampling given a fixed

number of monitors, or the number of monitors required for a given level of similarity between airsheds. We then demonstrate the new algorithm's application towards source apportionment of very large observation data sets, through the analysis of a year of Canada's hourly National Air Pollution Surveillance Program data, comprising 366,427 original observation vectors, a problem size that would be impossible with other source apportionment programs such as Positive Matrix Factorization.



## 1 Introduction

Air pollution is, globally, both an environmental and human health concern. The Global Burden of Disease project ranked air pollution the 4th leading cause of death for both males and females with a total of 6.67 million premature deaths in 2019 (Murray et al., 2020). Atmospheric transport of chemicals such as mercury, sulphur and nitrogen, and organic carbon can contribute to ecosystem impacts such as acidification in Canada's boreal forests (Makar et al., 2018). In agriculture, it has

also been estimated that ozone damage represents yield reductions of up to 7.2% globally for cereal crops (Tai et al., 2021). For these reasons, analysis of air pollution to characterise sources and inform control policies is an urgent and important matter. The causes of air pollution may be complex and require both observation networks of multiple chemical species and air-quality model estimates to determine sources and the effectiveness of emission control strategies.

Monitoring instrumentation and its operation may be expensive, necessitating evaluation of monitoring networks to

determine their effectiveness (The European Parliament and the Council of the European Union, 2004; European Parliament, Council of the European Union, 2008). Indeed, a recent review of Air-Quality Monitoring Network (AQMN) analysis papers found 77 different publications using a variety of modern and traditional statistical methods with a total of over 2000 citations between them (Verghese and Nema, 2022). Although most studies have focused on using monitoring network observations to conduct this analysis, air quality models may also be used as an aid in the process of

AQMN evaluation, by providing information about concentrations in regions where monitoring stations are not available. However, the large-scale analysis of air-quality model output to determine regions of similar chemistry is hampered by the poor scalability of existing analysis algorithms (Soares et al., 2018; Saxena et al., 2017).

Clustering is a well-known analytical technique with a long history in air quality research, which seeks to partition data into groups (or clusters) such that the within-cluster variance in the data is minimized while the between-cluster variance is

maximized. Here, the data may be, for example, time series, spectrograms, back-trajectory end-points, or temporal averages of observations of a suite of species, and may come from multiple fixed stations, measurement intensives, and/or air-quality models (Govender and Sivakumar, 2020). Hierarchical clustering is one of many possible approaches to clustering. Here, the algorithm proceeds stepwise, beginning with each data point in its own cluster, reducing the number of clusters at each analysis step by combining the two most similar remaining clusters. This similarity is scored using the dissimilarity metric,

which computes a "distance" between each cluster. Each step of the algorithm produces a partitioning (i.e., an assignment of every original data point to one of the remaining clusters) and the resulting series of partitionings may be visualised as a dendrogram, a tree-like diagram showing which clusters are combined at each step. Because of the depth of information that can be provided by these dendrograms, hierarchical clustering has been a particularly heavily-used method for AQMN analysis, often in combination with other clustering techniques such as principle component analysis (PCA; Stolz et al. 2020;

D'Urso et al. 2017). Two other popular clustering methods in the air quality field are positive matrix factorization (PMF) and K-means clustering.



Hierarchical clustering on larger datasets requires ever-increasing computational resources. The memory required scales as the number of input data squared, because the algorithm must initially compute and store the dissimilarity metric for each pair of data (without ordering) in the series, i.e., for clustering $n$ monitoring stations, there are $n$ possible choices for the first

station and $(n-1)$ choices for the second, making for $n^2 - n$ dissimilarity values to store. Worse, the computation time scales as the number of input data *cubed*, because the algorithm proceeds through $n$ steps, and at each step of the algorithm, we must first search out the smallest value in the size $O(n^2)$ dissimilarity matrix. For example, if clustering a dataset consisting of 100 time series (e.g., a $10 \times 10$ model gridded domain, or data from 100 measurement stations, n = 100) required 5 seconds on a given computer, a 10,000 timeseries dataset (i.e., 100x100 domain, or 10,000 measurement stations)

would require 5,000,000 seconds (or 1,389 hours). If the smaller dataset required 80kB of RAM to carry out these calculations, the larger dataset would require 800MB. Large data considerations thus may make analysis prohibitive, particularly as air-quality model output, with typical $(x, y)$ dimensions of hundreds of gridpoints in each direction ($x \times y$ total time series to cluster) covering a simulation region, are considered.

An improvement to the brute-force searching of the dissimilarity matrix (priority queuing) has been known for some time

and is used, for example, by a popular, publicly available python library, scipy.cluster.hierarchy (https://docs.scipy.org/doc/scipy/reference/generated/scipy.cluster.hierarchy.linkage.html). In short, rather than storing the dissimilarity matrix as a table and searching for the minimum remaining value at each clustering step, each row of the matrix is stored as in a data structure called a priority queue, or pqueue, which is a data structure that can return the lowest value (and its index) from a list in a single computation step (i.e., returning the minimum value is an $O(1)$ operation). This

improvement to the algorithm improves the average computation complexity of the overall hierarchical clustering algorithm from $O(n^3)$ to $O(n^2 \log(n))$ (Müllner, 2011). Even with this improvement, very large datasets may require excessive amounts of computational time, and there is still the matter of memory requirements. As a result, some problems may still bump up against reasonable limits to available computational resources for memory, processing time, or both.

To solve these problems, we turn to parallel computing. In particular, the Message Passing Interface (MPI) paradigm can be

used to distribute a single program across many different processors ("tasks" in MPI parlance) and even over separate computers ("nodes") in a high-performance computing environment (Pacheco and Malensek, 2022a). This allows the resources of many computers to be used at the same time on a single program. Additionally, we may apply the Open Multi-Processing (OpenMP) paradigm, using multiple "threads", where the threads represent many processor cores sharing memory access on a single node, for parallel execution (Pacheco and Malensek, 2022b). Careful use of these computational

models can result in significant decreases in processing time while increasing the memory available for processing by distributing the storage requirements across many threads in many tasks on many nodes.

There are several considerations in constructing a highly efficient parallel processing algorithm for very large datasets. A chief consideration is the time required for communication between processors ("overhead"). Depending on the architecture of a massively parallel computer system, this overhead may be smaller (e.g. between the processors within a node) or larger





(typically, between nodes). Communication within a node typically is best handled by the OpenMP parallization paradigm. Communication between nodes requires the MPI parallization paradigm. Due to the difference in communication times for these two paradigms, a possible strategy for code optimization is to carry out communication under OpenMP whereever possible, and MPI only when necessary – though the best strategy also incorporates as little communication between processors as possible. Therefore, the most benefit is provided by parallelizing "for loops" where each iteration is

independent of previous iterations.

Here, we present an implementation of the hierarchical clustering algorithm which, through the use of MPI and OpenMP parallelization, is suitable for analysing very large datasets on a high-performance computing cluster (e.g. ~300,000 individual time series derived from regional air-quality model output). We first will introduce the algorithm with a small example clustering analysis of a regional AQMN (the Wood Buffalo Environmental Association network; WBEA). We will

then describe the architecture of the program, and show we carefully constructed the hierarchical clustering code for optimal operation on massively parallel computers, analyze its memory and computational performance, and end with two example analyses: one making use of a year of regional air-quality model output (a 538x540 cell model output domain, i.e. 290,520 hourly concentration timeseries per chemical species clustered), and one on an entire year of hourly measurements taken from all stations of the Canadian National Air Pollution Surveillance (NAPS) air quality monitoring network (366,427

hourly measurements in all, where the clustering is across chemical species). The former shows how the new methodology may be employed as a new means for analyzing air-quality model output to aid in the design and optimization of AQMN, and the later demonstrates the use of HC for source-apportionment on a very large dataset. The associated Fortran90 code is provided open-source online at https://doi.org/10.5281/zenodo.8280445 (Lee et al., 2023). To our knowledge this is the first time a hierarchical clustering analysis has been performed on such large datasets; previous attempts having been limited by

the computational constraints in standard approaches as discussed above. The algorithm design makes possible the analysis of much larger datasets than had previously been possible using hierarchical clustering, or other techniques such as PMF or non-stochastic K-means.

## 2 Methods

### 2.1 Hierarchical clustering algorithm and AQMN analysis example

Hierarchical clustering is an algorithm for grouping data in a series of partitionings of progressively smaller size, such that the within-group variability is minimized while the between-group variability is maximized. Hierarchical clustering has been described in many other places but we will present a summary here for completeness. The metric chosen usually relates some property of the datasets to be compared, for example, Euclidean distance, Pearson's correlation coefficient (R), etc. The data could be any sort of 2-dimensional data: one dimension along which the dissimilarity metric is calculated, and the

other along which clustering is performed. For example, a sequence of time series from different monitoring stations to be compared through correlation would calculate the dissimilarity matrix as (1-R) between station time series (that is, time is



the dimension along which the dissimilarity matrix is calculated), with the subsequent clustering occurring across stations (the station identity becoming the axis for performing clustering). Initially, the dissimilarity is calculated between each possible pair of data (e.g. the correlation between all possible pairs of station time series). The clustering then proceeds

through a sequential examination of the dissimilarity matrix. At each step of this examination, the two remaining clusters with the lowest dissimilarity are identified, and their values of the dissimilarity metric are combined, reducing the number of total clusters remaining after the clustering step by one. The process of combining the two most similar clusters requires that the dissimilarity matrix be recomputed for all elements linking the new combined cluster and all the other clusters remaining at that step of the clustering. This recombination is referred to as the linkage step. Although there are many dissimilarity

metrics and linkages available(Murtagh and Contreras, 2017), in this paper we will use the (1-R) metric (Equation 1) and the average linkage (Equation 2) for all examples in this paper.

To provide a very simple example to illustrate the steps in the algorithm, using a full dissimilarity matrix (as opposed to the pqueue approach, for clarity): consider four monitoring stations with timeseries $S_1$ through $S_4$ . We first compute a $4 \times 4$ dissimilarity matrix $\mathbf{D}$ where each element $D_{i,j} = 1 - R_{i,j}$ where R is the Pearson correlation coefficient between the

timeseries $i$ and $j$. Initially we assign each timeseries to its own cluster of size 1, clusters C1 through C4. If the lowest value in the initial dissimilarity matrix were between $S_1$ and $S_2$, we add those series to a new cluster, C5, and remove C1 and C2 from the dissimilarity matrix. We now compute the dissimilarity scores between C5 and the remaining clusters C3 and C4, using the original dissimilarities between C1 and C3, and C2 and C3, and C1 and C4, and C2 and C4, i.e., $D_{5,3} = f\left(D_{1,3}, D_{2,3}\right)$ and $D_{5,4} = f\left(D_{1,4}, D_{2,4}\right)$, where $f$ is the chosen linkage function. Our dissimilarity matrix has now shrunk from

$(4 \times 4)$ to $(3 \times 3)$ in this first stage of clustering, because we removed 2 clusters (C1 and C2) and added 1 (C5). The process now repeats for the remaining three clusters until the final two clusters are combined into one cluster which has combined all the elements.





**Figure 1 A dendrogram of an air quality monitoring network (top panel) based on the (1-R) metric for SO₂ concentrations and two possible partitionings based on that dendrogram, one at a dissimilarity level of 0.4 (R = 0.6) (bottom-left panel, blue dot-dashed line in top panel) and the other at a dissimilarity level of 0.7 (R=0.3) (bottom-right panel, orange dashed line in top panel). In the bottom panels, all stations in a given group are represented by the same colour and symbol. Made with Natural Earth.**






In order to further orient the reader, we present an example of how the algorithm can be used on an AQMN. For this example, we clustered hourly continuous $SO_2$ measurement timeseries from 21 stations in the Wood Buffalo Environmental

Association (WBEA) monitoring network in Alberta, Canada, for the period January 1, 2022 to December 31, 2022 (Figure 1). The clustering was performed with an initial correlation dissimilarity metric $D = (1 - R)$, where R is the Pearson correlation coefficient computed between hourly timeseries at two stations. The correlation dissimilarity is shown in equation (2) where $x_t^i$ is the $SO_2$ concentration at station $i$ at time $t$ and $T$ is the total number of hourly measurements available. We also note that the resulting matrix **D** is symmetric ($D_{i,j} = D_{j,i}$); the memory space storage requirements may be

reduced by taking the symmetry into account.

$$D_{i,j} = 1 - \frac{\sum_{t=1}^{T}(x_t^i - \bar{x}^i)(x_t^j - \bar{x}^j)}{\sqrt{\sum_{t=1}^{T}\left(x_t^i - \bar{x}^i\right)^2}\sqrt{\sum_{t=1}^{T}\left(x_t^j - \bar{x}^j\right)^2}} \tag{1}$$

We applied the average linkage for computing D (equation (2) between the newly combined cluster $ij$ at each step and the other remaining clusters, $k$. Once again, this is intended as an example of one of the 7 linkages available. Common linkages include the minimum linkage, which uses the minimum dissimilarity of all the pairwise elements of each cluster being combined, or the average linkage, which uses the average of the dissimilarities between the members of the clusters. The

revised matrix where entries have been linked is given in this case by:

$$D_{ij,k} = \frac{c_i \cdot D_{i,k} + c_j \cdot D_{j,k}}{c_i + c_j} \tag{2}$$

Where $i$ and $j$ in (2) represent the entries in the previous iteration of the algorithm that are being combined, and $c_i$ is the number of items in cluster $i$ in the previous step and $c_j$ is the number of items in cluster $j$ in the previous step, and the new cluster will have been generated from $c_{ij} = c_i + c_j$ original items. Initially, $c_i = 1 \; \forall \; i$ since each item is placed in its own cluster at the beginning of the process. After each step, all elements for $i$ and $j$ have been removed from the dissimilarity

matrix, while new elements for $ij$ have been added, for a net decrease of $1 \times 1$ in the size of the dissimilarity matrix.

The clustering output may be represented as a dendrogram, shown in the top panel of Figure 1. The top panel also shows two (arbitrary) values of the dissimilarity metric D = 0.4 and D= 0.7 (which, in this case, correspond to values of R of 0.6 and 0.3, respectively). For the smaller value of the dissimilarity metric (the higher correlation between stations), the dendrogram shows that there are 16 clusters of stations at R=0.6 (these are, where brackets {} indicate station records which have been

clustered together, {AMS17, AMS19}, AMS23, AMS30, {AMS09, AMS13, AMS01, AMS25},AMS02, AMS04, AMS11, AMS05, AMS26, AMS20, AMS21, AMS27, {AMS06, AMS07}, AMS14, AMS22 and AMS29). At the higher level cut of the dendrogram, 11 clusters may be seen, corresponding to R=0.3 ({AMS17, AMS19}, {AMS23, AMS30, AMS09, AMS13, AMS01, AMS25}, AMS02, AMS04, AMS11, AMS05, AMS26, AMS20, {AMS21, AMS27}, {AMS06, AMS07}, and {AMS14, AMS22, AMS29}). These clusters are also presented in tabular form in Table 1. The clustering thus identifies

groups of stations which are grouped together based on a specific value of the dissimilarity matrix shown. Stations which



join a cluster at the bottom of the dendrogram are the least dissimilar (most similar; when using the (1-R) metric, these are the stations with time series that have the highest correlation). Stations which join a cluster at the top of the dendrogram are the most dissimilar from the others; these are the stations which have the lowest correlation to other stations' time series. In this particular example, stations AMS01 and AMS25 are the most similar, while station AMS05 is the least similar, to the

other stations in the dataset examined.

**Table 1 Clustering analysis of SO₂ concentrations from the Wood Buffalo Environmental Association air quality monitoring network at two dissimilarity cutoff levels with the (1-R) metric. Each station (AMSXX) begins in its own cluster and stations are grouped together at progressively higher dissimilarity levels (i.e., lower correlation coefficient values).**

| Cluster | Members with $R \geq 0.6$ | Members with $R \geq 0.3$ |
|---------|---------------------------|---------------------------|
| 1 | AMS17, AMS19 | AMS17, AMS19 |
| 2 | AMS23 | AMS23, ASM30, AMS09, AMS13, AMS01, AMS25 |
| 3 | AMS30 | AMS02 |
| 4 | AMS09, AMS13, AMS01, AMS25 | AMS04 |
| 5 | AMS02 | AMS11 |
| 6 | AMS04 | AMS05 |
| 7 | AMS11 | AMS26 |
| 8 | AMS05 | AMS20 |
| 9 | AMS26 | AMS21, AMS27 |
| 10 | AMS20 | AMS07, AMS07 |
| 11 | AMS21 | AMS14, AMS22, AMS29 |
| 12 | AMS27 | |
| 13 | AMS06, AMS07 | |
| 14 | AMS14 | |
| 15 | AMS22 | |
| 16 | AMS29 | |





A given level of the D metric thus defines how stations may be grouped together. The spatial locations of these groupings may be shown on a map (bottom panels of Figure 1). Choosing different symbols for each cluster within the two values of D in the upper panel allows the reader to see the spatial relationship between the different station records: at increasingly higher levels of dissimilarity (increasingly lower levels of R in this case), more stations group together. Here, for example, it can be seen that stations AMS9, AMS13, AMS01, and AMS25 (red squares, Figure 1b) are almost co-located and hence the

clustering at R=0.6 is reasonable, as is the clustering of stations AMS06, AMS07). At a lower level of correlation (Fig. 1c), stations AMS14, AMS22 and AMS29 all fall into a cluster in the lower right of the panel.

**2.2 Scalable implementation of hierarchical clustering software**

While the above example demonstrates hierarchical clustering for a simple case of 21 records being compared and clustered, more careful algorithm design considerations come into play when larger datasets are to be considered. We discuss these

design issues, key to our revised algorithm, in this section.

As with hierarchical clustering algorithms in general, our implementation begins by computing the dissimilarity matrix. As noted above, it is not actually necessary to store the entire dissimilarity matrix because $D_{i,j} = D_{j,i}$ which means the matrix is symmetric and we need only store the upper or lower triangle of the entries of the matrix. Also, in order to avoid searching through an entire matrix for the lowest value at each step (described below) we follow Müllner (2011) and store each row of

the matrix as a priority queue (pqueue), a data structure which returns the lowest value and its position in O(1) time, i.e., returning the lowest value always takes a constant amount of computation time, no matter how many elements are currently in the pqueue. This speeds the average time of the algorithm up from $O(n^3)$ to $O(n^2 \ln(n))$.

After computing the initial dissimilarity matrix and storing it in pqueues, the algorithm then proceeds iteratively. At each step, we search for the minimum value in the matrix, $\left( i_{min}, j_{min} = \underset{i,j}{\mathrm{argmin}}\, \mathrm{D} \right)$, and cluster $i_{min}$ with $j_{min}$, creating a new

cluster $k$ which contains all the members of $i$ and $j$. The dissimilarity between the new combined cluster $k$ and all the other clusters not in $k$ is not computed explicitly; rather, a linkage operation as defined above is performed for this combination. The algorithm repeats stepwise until there is only one cluster remaining which has linked all the original items. The output of the algorithm is the list of the clusters combined, and at what dissimilarity they were combined, at each step of the clustering. From this output, one can create the dendrogram as shown above. These dendrograms can be useful analysis tools for finding

efficient or highly-correlated clusters and quickly spotting outliers in the dataset. For very large datasets, dendrograms become more difficult to interpret, and spatial maps such as Figure 1 (b,c) become more useful for interpretation of the clustering analysis.

The algorithm also requires $\propto \frac{n^2}{2}$ elements of memory allocation in order to store the dissimilarity matrix. In algorithm design, it is generally possible to trade computation time for memory – values can be computed once and stored for later

access, or recomputed when needed multiple times, which is generally slower than reading from memory. Given the





computation complexity of computing the dissimilarities, and the large amount of fast access memory available in current computing systems, the latter approach would not be a prudent strategy. Instead, we here increase the available memory by applying MPI parallelization, which gives the algorithm access to many nodes, and therefore more memory.

In this study we used Intel(R) MPI Library for Linux* OS, Version 2021.5 and OpenMP 4.5 in the Intel Fortran Compiler
version 2021.5.0. The code architecture was designed to allow for the computation and memory burdens to be dynamically allocated across multiple processers at runtime. OpenMP loop parallelization directives were added where appropriate, to allow some loops to be computed in parallel on multiple processor cores on a single node, which in many instances is faster than MPI parallelization because of the decreased need for inter-task communication. A more detailed description of the considerations made in designing the high-performance version of the algorithm is presented in Appendix A.

### 230  2.3 High-performance computing platform

In the examples which follow, we used a high-performance computing cluster operated by Shared Services Canada. Each node has two Intel® Xeon® Platinum 8830 CPUs running at a clock speed of 2.30 GHz, which provide 80 cores total. Each node has 256 GB of memory available to those 80 cores. There are 358 nodes available for a total of 28640 total cores and 89.5 TB of memory. The nodes are connected to one another and to the network file system by InfiniBand HDR network.

In order to examine the performance statistics of the software as written on the platform available, we performed an additional test where the program was constrained to a single core. This limited the maximum size of the problem it was possible to compute, but allowed for comparable statistics between problems of different sizes, without the concerns of interfering effects from the parallelization overhead, to be generated.

### 2.4 Clustering examples

To demonstrate the use of our new high-performance clustering software, in this study we used two different types of data in our examples. In the first example, we used hourly time series of individual air quality pollutants concentrations and deposition fluxes, generated by an air-quality model on a rectangular grid. In the second example, we perform factor analysis on every hourly set of observations performed across Canada's National Air Pollution Surveillance (NAPS) network in the year 2019.

### 245  2.4.1 Clustering Example 1: Concentration timeseries from a regional air quality model

As a first demonstration of the capabilities of the parallelized software on a high-performance computing platform, we chose the output from a 15-month simulation of a regional air quality model, Environment and Climate Change Canada's (ECCC) Global Environmental Multiscale-Modelling Air-Quality and CHemistry (GEM-MACH) model. GEM-MACH is ECCC's main regional air-quality model, designed for regional pollution forecasting, and policy emissions scenario simulations. The
model operates on a nested, rotated, latitude-longitude domain, here making use of a 10km grid cell size outermost domain covering North America, nested down to a 2.5km grid cell size inner domain (538x540 grid cells) covering the Canadian





provinces of Alberta and Saskatchewan (Moran et al., 2010; Makar et al., 2018; Moran et al., 2018). The inner grid, the 2.5km Oil Sands Domain, from which the input to the clustering algorithm was taken, is shown in Figure 2 as the smaller blue rectangle, while the 10km regional domain within which this 2.5km domain is nested, is shown as the large green
rectangle. The model employs a terrain-following hybrid coordinate system with 84 vertical levels. The model is an "online" system, meaning the meteorological and chemical predictions are carried out in the same system, here "fully coupled" to include the aerosol radiative direct and indirect feedback effects (Makar et al., 2015a; Gong et al., 2015; Makar et al., 2015b). The model run was based on GEM-MACH version 3.1.0b.2, with several updates: the SAPRC11 gas-phase chemistry mechanism (Carter and Heo, 2013), inorganic heterogeneous chemistry (Miller et al, 2023, submitted), plume rise
incorporating the effects of latent heat release of stack exhaust water (Fathi et al, 2023, submitted), updated cloud scavenging (Ghahreman et al, 2023, submitted), online photolysis (Majdzadeh et al., 2022), semi-Lagrangian advection (Yeh et al., 2002),  finite difference vertical diffusion with area source emissions applied as a lower boundary condition on the diffusion equation, and emissions from National Pollutant Release Inventory (NPRI) and Air Pollutant Emission Inventory (APEI) as well as ground, aircraft, and satellite-based data sources (Zhang et al., 2018; McLinden et al., 2020). The emissions
were based on the year 2015 reported emissions and updated with observed and projected emissions to 2018. The output from the model's 2.5km domain was used as the input data to the hierarchical clustering algorithm, and consists of hourly concentration or deposition values on a 540 by 538 grid, making for an input dataset size of 290,520 timeseries for each individual species. Soares et al. (2018) demonstrated that hierarchical clustering could be used to analyze regional air quality model output, with clusters of similar model grid-cells defining sub-regions in the model domain of similar chemical history
("airsheds"). However, that work was only able to handle relatively small numbers of model grid cells due to being limited to a single CPU core and the memory available on a single compute node. Here we make use of our new clustering code optimized for parallel processing in a high-performance computing facility, demonstrating the first use of clustering to analyse the output for an entire regional model.



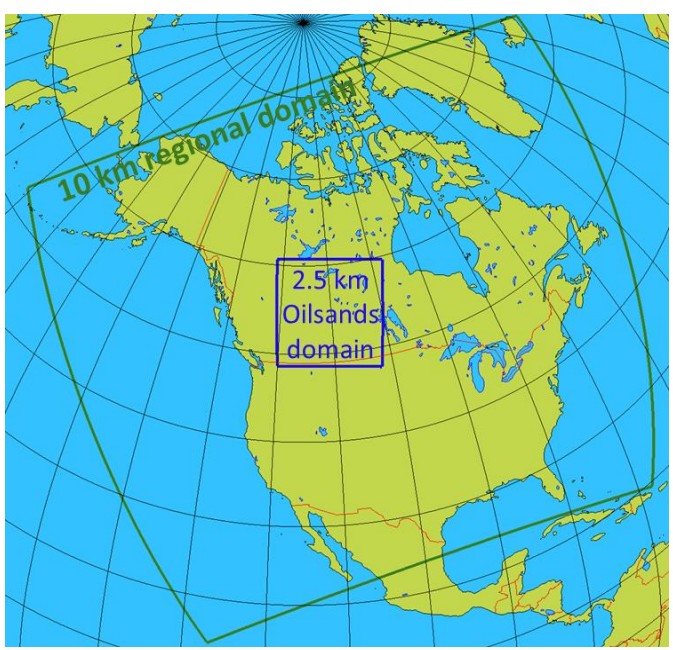

**Figure 2 The nested model domains from the GEM-MACH simulation used in Example 1 of the clustering algorithm. The large green box denotes the 10km regional domain covering all of North America while the smaller blue box shows the extent of the 2.5km-resolution nested domain covering the Canadian provinces of Alberta and Saskatchewan.**

**2.4.2 Clustering Example 2: Factor analysis on air quality monitoring network data**

As another demonstration of the scalability of this program, we perform receptor modelling or factor analysis on an entire
year of hourly data from Canada's NAPS network. The NAPS network has been taking air quality measurements since 1969 and has data from 260 stations in 150 communities. For this study, we used the NAPS continuous data which reports hourly values for NO, $NO_2$, $NO_x$, $SO_2$, $O_3$, CO, $PM_{2.5}$ and $PM_{10}$. Due to the COVID-19 pandemic, there were data continuity issues starting in 2020, so for this study we used the year 2019. Additionally, many stations report all pollutants listed above except $PM_{10}$. Therefore, we only used the remaining species for the receptor-modeling analysis. We filtered the 2019 data for hours
and locations which included values for all 7 species. Each resulting hourly concentration set of 7 species (across all available stations and times) was then treated as an independent data point for clustering, leaving us with 366,427 data points from 51 different stations. In order to perform hierarchical clustering in a fashion treating each species as equally important, we normalized each species' concentration independently of one another, subtracting from each concentration the lowest observed value for that species (at any station) and dividing by the difference between the highest and lowest values for that
species (at any station), giving each species a range from 0 to 1. The dissimilarity matrix was computed as $(1 - R)$ again, but in this case between the sets of pollutant concentrations, which we then clustered in the spatio-temporal dimension. That is, we used equation (1) as the metric again, but in this case the index $t$ is the species out of the $T = 7$ species we used and the index $i$ and $j$ are the spatio-temporal indices. In this way, each set of 7 concentrations measured at an hour at a NAPS

 

station can be assigned to a cluster. The clustering thus agglomerates the relative magnitudes of the different species across

stations and times, with the intent of determining the extent to which similar sources and chemistry have influenced the sites. This example is intended to show that the software is flexible enough to handle a very different types of data, while demonstrating how the algorithm's parallelization allows us to tackle problems with hundreds of thousands of initial datapoints, which would be prohibitive using other factor analysis techniques such as PMF, or K-means, without requiring special treatment or non-deterministic searches (Bahmani et al., 2012).

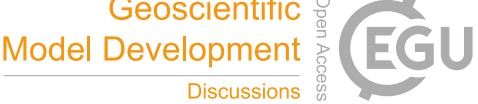

## 3 Results

### 3.1 Software performance

Using the previous serial version of the algorithm with the hardware available at the time, it was not possible to cluster more than approximately $100 \times 100$ or 10,000 or approximately 3% of the model domain within the wallclock time limits imposed on that system (4.5 hours; Soares et al. 2018). The new software improves upon the old architecture in 2 ways: replacing the dissimilarity matrix with a list of pqueues, and parallelization with the MPI and OpenMPI APIs. In order to test the improvement associated with switching to pqueues, we also carried out simulations where we constrained the new program to a single CPU and restricted OpenMP threads to 1, meaning the code was executing entirely serially. Figure 3 shows the timing curve of the new hierarchical clustering software versus problem size on a single thread (i.e. a single computing core), carried out using this constrained setup, to demonstrate the performance of the program *without* parallel processing assistance to reduce processing time. The best fit for the timing curve, in seconds, was given by $1.92 \times 10^{-7} x^2 \ln(x)$ ($R^2 = 1.000$), where $x$ is the number of points being clustered. The figure also shows the timing curve for the old code, which increases much more quickly as the domain size increases, leading to intractable compute timescales with fairly limited problem sizes.

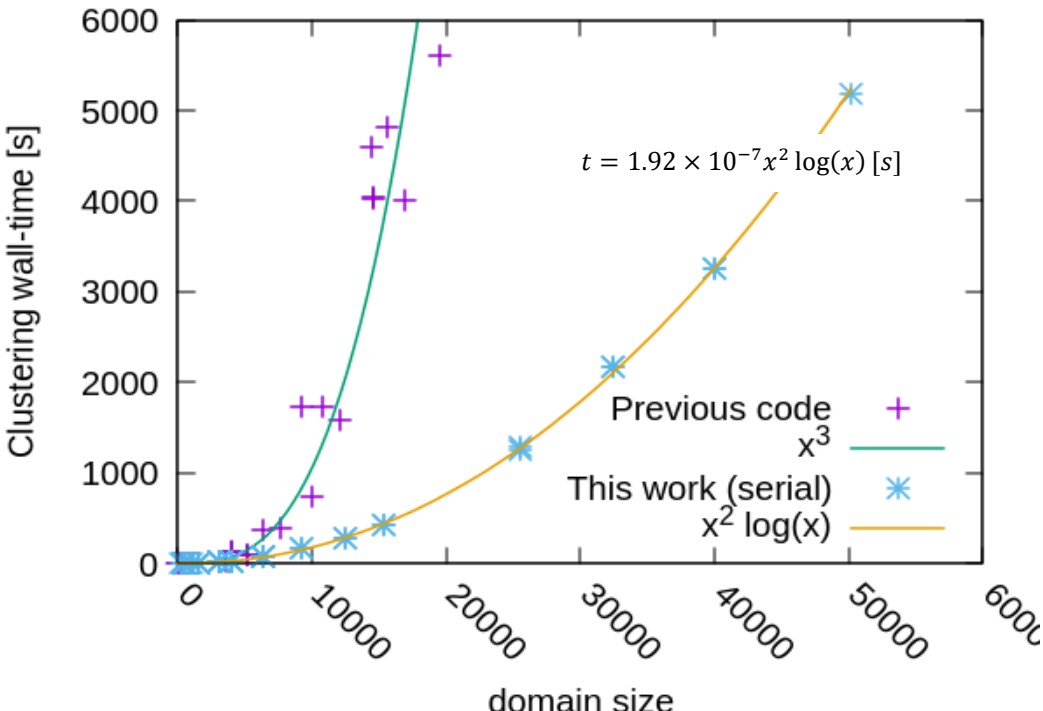

**Figure 3 Clustering time (in seconds) for two versions of the hierarchical clustering program constrained to a single MPI task and a single OpenMP thread versus domain size with best fit curve shown. A lower time is better. Each marker on the "This work" curve represents between 4 and 10 tests of that problem size but the execution times were very consistent and showed so little spread as to be indistiguishable on the plot.**





The second improvement we made to the software was the use of parallelization. This actually serves two purposes: to decrease the wallclock time by allowing multiple processors to compute in parallel, and to increase the amount of memory available to the system as a whole and break the dissimilarity matrix storage up over many nodes. The parallelization provides a modest improvement in clustering time – we were able to decrease the time to cluster the entire model domain of $538 \times 540$ for a given tracer or deposition variable down from a hypothetical 24 hours (by extrapolating the above timing curve for a single processor) to an average of 12 hours, 37 minutes of wallclock time (range: 12:33 to 12:41) on 100 full

nodes with 10 MPI tasks and 8 OpenMP threads on each node. The key improvement aside from the factor of two reduction in processing time with the use of parallel processing is the capability of the software to use larger amounts of computer memory, enabling the analysis of larger problems.

**Table 2 Clustering memory (selected problem sizes) usage versus problem size for single-thread execution**

| Problem Size | Memory [GB] (stdev in brackets) |
|---|---|
| 1024 | 0.177 (0.00403) |
| 3136 | 0.57 (0.00255) |
| 4096 | 0.888 (0.00461) |
| 9216 | 3.75 (0.00416) |
| 12544 | 6.79 (0.00391) |
| 29241 | 35.7 (0.00694) |
| 38416 | 61.2 (0.0119) |
| 44944 | 83.4 (0.0272) |
| 53824 | 120 (0.0317) |
| 58081 | 140 (0.21) |

The results for memory usage versus problem size (i.e., number of initial clusters) are show in Table 2. These are provided mainly for interest, as the values follow almost perfectly the expected $x^2$ curve with a constant of approximately 40bytes per domain-size-squared. This information can help guide users in choosing how many MPI nodes/tasks to parallelize over, depending on the memory availability of their own compute cluster.

**3.2 Clustering Example 1: Concentration timeseries from a regional air quality model**

Figure 4 shows the maps generated by clustering the (1-R) metric of 4 common air-quality species, $PM_{2.5}$, $SO_2$, $NO_2$ and $O_3$ at the (1-R)=0.75 cutoff threshold, except for $O_3$, which shows the 0.5 cutoff because the system is reduced to a single





cluster at (1-R) = 0.65. There are notable differences in the maps, the first of which is the number of clusters remaining at the same dissimilarity score cutoff: for PM$_{2.5}$ there are 154 clusters remaining, for SO$_2$ 124, and for NO$_2$ 166, while for O$_3$ there are only 11 clusters at the 0.5 cutoff.

**Figure 4. Cluster maps with (1-R) dissimilarity cutoffs for PM2.5 at 0.75 (top-left), SO2 at 0.85 (top-right), NO2 at 0.8 (bottom-left) and O3 at 0.5 (bottom right). Colour maps are arbitrary and adjacent colours are not indicative of any relationship. Made with Natural Earth.**

This example shows how a user of the code could use it to determine airsheds within which concentrations of each species would correlate at a given level – and that the size and location of the airsheds vary, depending on the chemical species. The differences reflect the extent to which the four species are influenced by different emissions, deposition, transport, and chemical factors. PM$_{2.5}$ for example (Fig. 4 a) is influenced by both surface sources (fugitive dust), elevated sources (stacks,





forest fires), and secondary chemistry – consequently, at the 0.75 cutoff level, a large number of clustering regions are identified for this species. $SO_2$ is more influenced by large stack sources and subsequent downwind chemistry and deposition, resulting both in larger clusters and a smaller total number of clusters compared to PM2.5. $NO_2$ is influenced by both surface, stack and biogenic sources, as well as by transformation to end products like $HNO_3$ and by deposition, resulting in the highest number / and smallest spatial extent of clusters of the chemicals examined here. The very small number of

clusters for $O_3$ in this region at a relatively smaller value of (1-R) shows that most of the factors influencing $O_3$ are regional in nature, aside for a few clusters near large sources of NOx emissions such as the Oil Sands, a major oil-producing region in Alberta, Canada, marked as Fort McKay on the maps. There $O_3$ is influenced largely by $NO_x$ titration where freshly emitted NO removes ozone. Figure 4 d also shows the influence of elevation (Rocky Mountains, red cluster on the left of the panel), and differences in deposition between boreal forest (brown cluster) and prairie / grassland vegetation types (pink cluster).

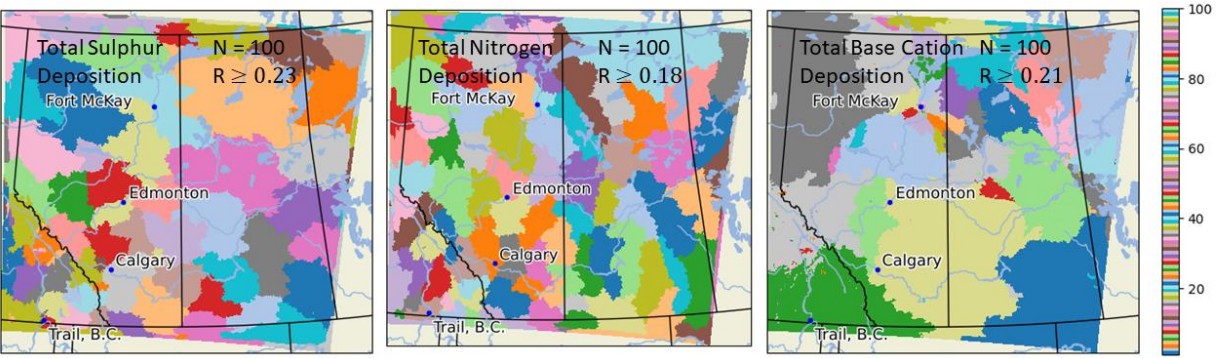


**Figure 5. 100-cluster maps for total sulphate (left), total nitrate (centre), and total base cation (right) for the entire 540 x 538 pixel model domain. Colour scale is arbitrary; each contiguous region of one colour represents one cluster. Provincial, state, and federal borders are marked in solid black. Most AOSR operations occur within about 50 km North, Northeast, and South of Fort McKay. Made with Natural Earth.**





Figure 5 shows an alternative form of model analysis, this time for three sets of *deposited* quantities, the flux to the surface of total sulphur, total nitrogen, and total base cations. Sulphur and nitrogen deposition may lead to ecosystem acidification and potential ecosystem damage, while base cation neutralization may offset that potential damage. In this instance the maps depict the airsheds for deposition where 100 clusters is used in each case. This form of analysis can be used to define regions of equal representativeness, for example, as advice in planning a 100 station monitoring network for each species

(that is, a station placed within any one of the coloured regions would be representative to the correlation level for that number of stations). Each of the three maps tells a very different story. While the 100 clusters are spread fairly evenly across the region in the sulphate and nitrate maps, the base cation map shows around 20 fairly large clusters that cover the model domain with the other 84 clusters being difficult to discern at the scale of the map, including 12 clusters which contain only a single unique grid cell, very close to sources of primary particulate matter, the main source of base cations. The clustering

analysis thus allows the user to discern the relative influence of local sources on local versus regional deposition.

### 3.3 Cluster Example 2: Factor analysis on air quality monitoring network data

After applying the clustering algorithm to the entire filtered 2019 hourly NAPS dataset as described above, we arbitrarily chose a cut-point of 50 clusters. The number assigned to each cluster does not provide any particular meaning, except that within a given cluster, all observations are related within a certain R-value ($R \geq 0.90$). The ordering of the cluster numbers

(1-50) is not representative of any further relationship – the data in cluster 1 is not necessarily any more similar to the data in cluster 2 than it is to the data in cluster 3 or cluster 50. The approach is analogous to positive matrix factorization; certain clusters by virtue of where / when they occur, may describe similar aspects of the air quality occurring at specific stations. Figure 6 shows a histogram of the occurrence of each cluster across all 51 sites in our filtered NAPS network data. The cluster which appeared the most frequently was cluster 45, into which 225,718 data points (out of 366,427 or 62%) samples

were clustered. Cluster 44 had the next-highest number with 61,652 data points (17%), while the next 3 clusters (49, 47, and 46) had 31,160, 24,699 and 13,317 samples. Following this were two clusters (48 and 40) with 4,111 and 2,089 samples. There were another 11 clusters whose counts ranged from 117 to 639, and the remaining 32 clusters had fewer than 100 samples each, 7 of which had only 1 sample each, meaning that clusters only showed up at one timepoint at one station, and are therefore indicative of relatively rare events.



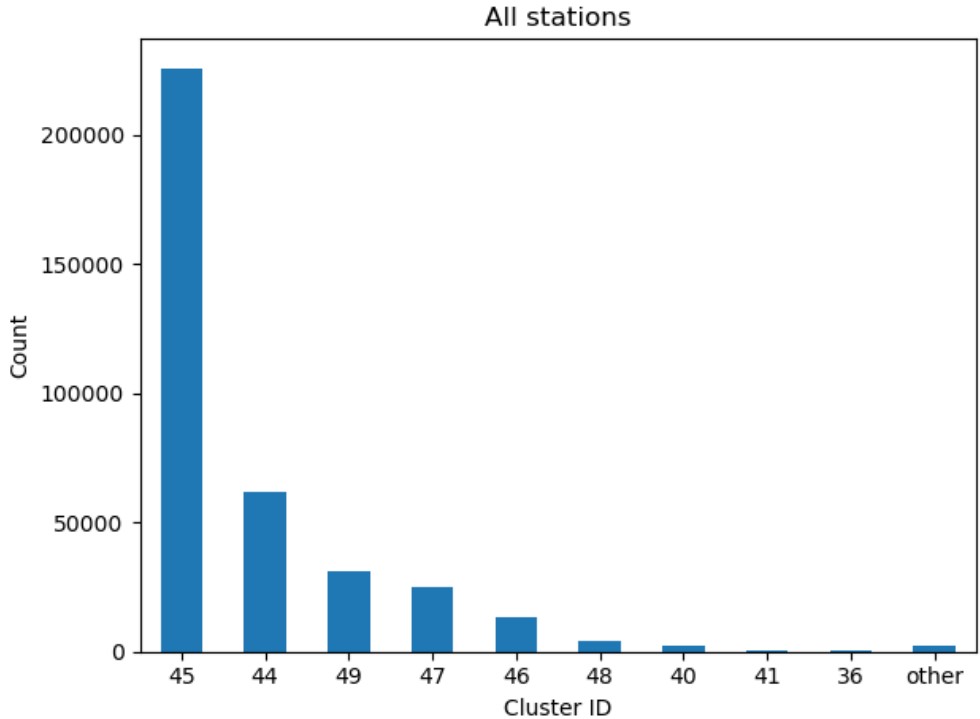


**Figure 6 Histogram of frequency of 50 clusters across all NAPS sites for all hourly 2019 data.**

We show one means by which the results of the clustering may provide information on pollutant concentrations and potentially sources, in the form of maps in which each the frequency for which each of the 50 clusters appeared at a given

station. Figures 7, 8 and 9 show these as inset pie charts on a map of the station locations. The largest segment in every pie chart but one is 45, indicating that this cluster is the most prevalent cluster at all sites, by a wide margin in all cases except two Metro Vancouver sites. At those two sites, 44 is nearly as frequent as 45, and at one of those two sites, 46 and 49 are even more frequent than 44. At only one site, Winnipeg, cluster 45 (the most common cluster *at all other sites*) is the 3rd most common cluster, after clusters 44 and 49.





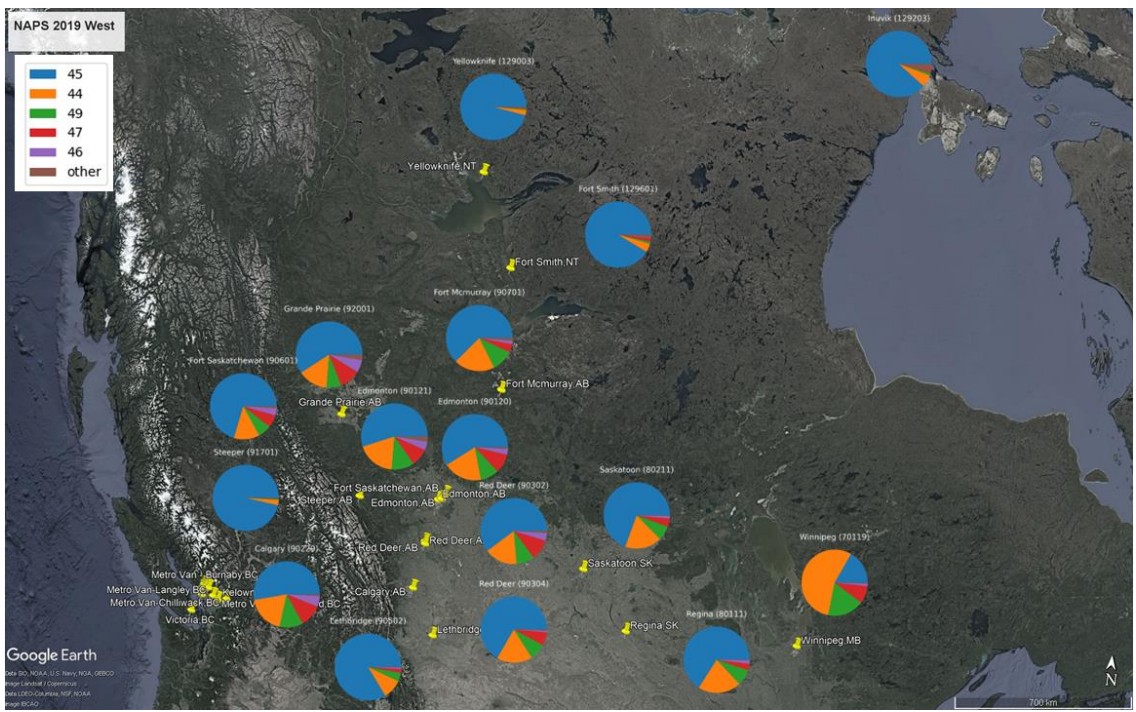

**Figure 7 Histograms of the frequqency of appearance of the 50 clusters at Western-Canada NAPS stations. The yellow pins indicate the location of a NAPS station whose historgram is shown in a nearby inset. Map data: © Google Earth, Landsat/Copernicus.**



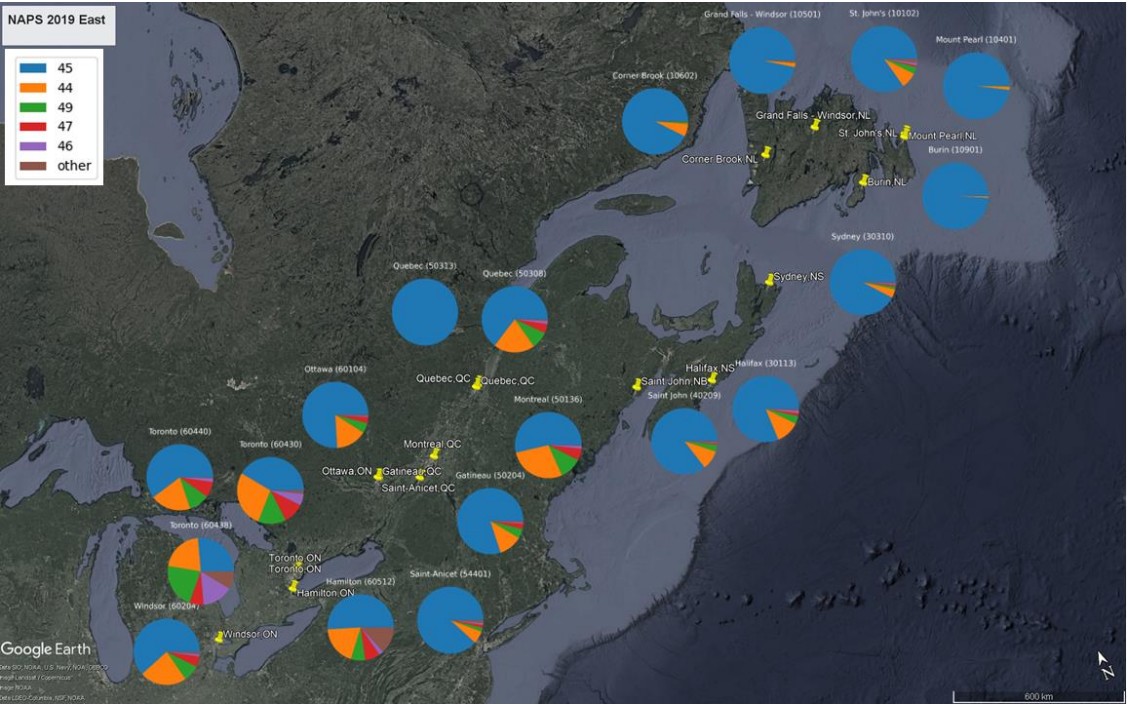

**Figure 8 Histograms of the frequqency of appearance of the 50 clusters at Eastern-Canada NAPS stations. The yellow pins indicate the location of a NAPS station whose historgram is shown in a nearby inset. Map data: © Google Earth, Landsat/Copernicus.**





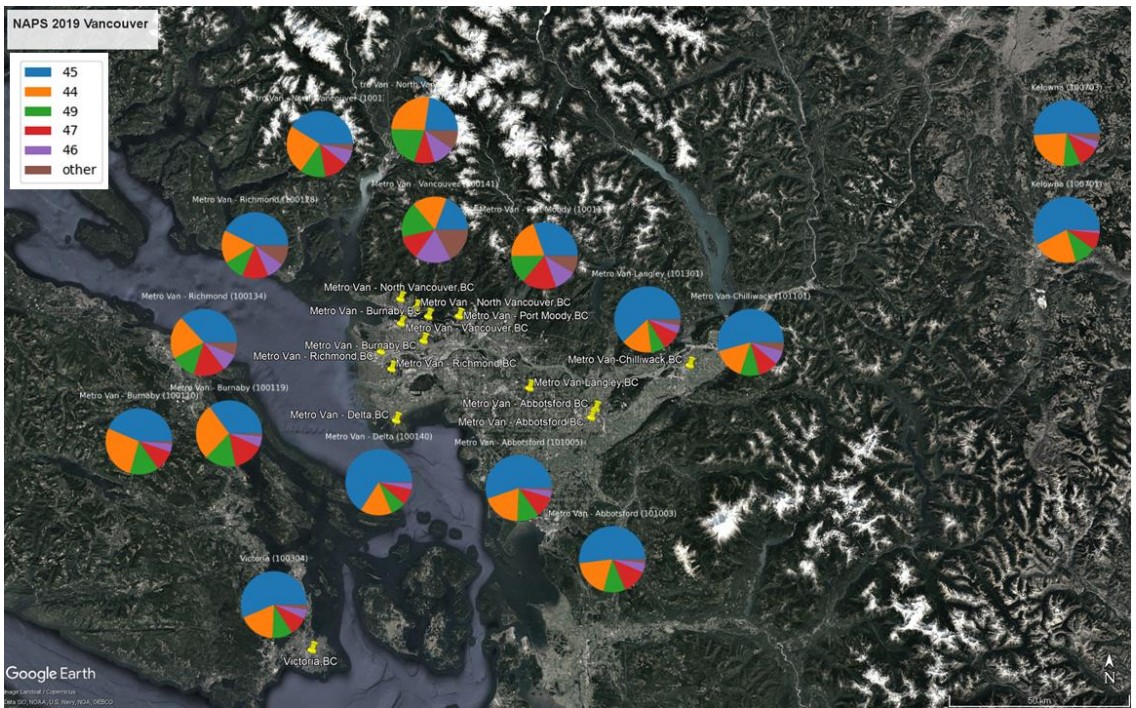

**Figure 9 Histograms of the frequqency of appearance of the 50 clusters at Vancouver and Victoria NAPS stations. The yellow pins**
**indicate the location of a NAPS station whose historgram is shown in a nearby inset. Map data: © Google Earth,**
**Landsat/Copernicus.**

The frequency of clusters appearing at individual stations may then be linked to potential source information, in a manner
similar to the identification of components associated specific sources in Principal Component Analysis. Figure 10 shows
the concentration profiles for the first 14 clusters ranked by frequency of occurrence. The most common cluster, 45, shows
the lowest values for CO, $PM_{2.5}$, NO, $NO_2$, $NO_x$ , while $O_3$ is the highest for this cluster. Given that many of the stations are
within urban areas, this cluster likely represents "background" air, with low pollutant concentrations, and minimal titration
of ozone by NO. The next 6 clusters in order of frequency show increasing concentrations of CO, $PM_{2.5}$, NO, $NO_2$, NOx and
decreasing $O_3$, indicating increasingly polluted regimes where ozone is being removed by NOx titration. Interestingly, for the
first 9 clusters (all 7 in the top panel of Figure 10 and the first 2 of the bottom panel) $SO_2$ shows no difference between
clusters, and has large within-cluster variance, so $SO_2$ differences are not an important contributor to those clusters.





**Figure 10 Concentration profiles for the 14 most common clusters across all NAPS stations for the year of 2019. The top panel shows the 7 most common, the bottom panel shows the following 7 most common. The Y axis is in ppb for all species except PM$_{2.5}$ which is ug/m3 and CO and SO2 which have been scaled to fit the axis. The error bars indicate 1-standard deviation from the mean for that cluster.**

The 10th, 12th, 13th and 14th most common clusters (clusters 43, 42, 39 and 17), on the other hand, are driven largely by their SO$_2$ differences. These 4 clusters appeared a total of 1,073 times at 19 different stations as spatially distant as Metro





Vancouver – Burnaby, BC (station 100110) and St. John's, Newfoundland (station 10102) which are separated by 5000km. These clusters appeared most frequently at Hamilton (station 60512; 797 occurrences) and Edmonton (station 90121; 93 occurrences). The last of these, cluster 17, is characterised by the highest average $SO_2$ concentration of 48.6 ppb. This cluster appears 152 times in total – it occurs most frequently at Windsor (60204; 149 occurences) while also appearing at Edmonton twice and Sydney once. Noting that $SO_2$ is mostly emitted by large stack sources, the most likely explanation for these clusters is that they are being impacted from a distinctive class of $SO_2$ source, e.g., plumes from powerplants, petrochemical facilities, ocean-going vessels, etc., which comprise the dominant form of $SO_2$ emissions.

Further analyses of both of these datasets are of course possible – the aim here was to demonstrate the use of the new clustering algorithm on complex air-quality data, and that the results provide fresh insights into sources, pollutant transformation, and deposition.

## 4 Discussion

We expect that users of the new algorithm will wish to spend some time testing different configurations of number of processors and the relative MPI/OpenMp breakdown in order to find the most efficient MPI/OpenMP configuration for their system, due to differences in high performance computing architecture. Some of these differences may occur at specific problem sizes, and are idiosyncratic to a given architecture. As an example, during some tests we carried out early on during development (not shown in the results), we noted a change in the relationship between clustering time and problem size for a single thread at a problem size of around 50,000 initial clusters using only a single thread, which could be an indication of a possible hardware limitation. Below this threshold, we see the expected $O(n^2 \log n)$ behaviour, but above the slope becomes much steeper. Our current hypothesis is that as the amount of RAM accessed by a single thread increases past a certain threshold on our system, the RAM access time increases. The location of the change in relationship corresponds with a maximum memory usage of around 120GB which is approximately half the total available RAM for the whole 80-core node, which lends support to this hypothesis. The authors would like to emphasize that there may be many considerations required to obtain the best performance on any given high-performance computing cluster.

In our example clustering maps in Figure 4, it is clear that the choice of similarity threshold has an important influence on the size and number of airshed domains that will result from clustering, as does the specific species for which clustering is carried out. Ozone, which in the region depicted in Figure 4(d) is driven largely by the diurnal cycle in incident solar shortwave radiation for diurnal formation and land surface type for depositional loss, as well as titration within regions of fresh NOx emissions, has very large airsheds, which occur at a high similarity throughout the model domain. Sulphur dioxide and nitrogen dioxide are more dependent on local emissions, resulting in much smaller airsheds, especially around large emissions sources. One notable example is the unique cluster located at the very southern border of British Columbia near the 3-way border of Washington, Idaho and BC, shown in orange on the $SO_2$ map in Figure 4(d). This cluster is likely driven by the Teck Cominco Smelter, the largest of its kind in the world, located at Trail, BC (Montgomery, 2009).





Similarly, the larger number of clusters of small spatial extent around Fort McKay (which is surrounded by the Canadian Oil Sands surface mining facilities) for $PM_{2.5}$, $SO_2$, $NO_2$ and $O_3$ (Figure 4), is a result of local sources contributing to concentrations, or in the case of ozone, titration by $NO_x$. This highlights the value of being able to perform this clustering
algorithm over such a large dataset; using only air quality model simulation output, the algorithm is able to capture the area-of-influence of these sources, in a way that would normally require an expensive field campaign with many measurement locations.

This type of model analysis can be useful in both air quality studies and model development. If we define an airshed as a geographical area that is impacted by similar sources and meteorology, we can use the clustering algorithm to find all the
airsheds for a given level of similarity or a given total number of airsheds. This allows the user to improve the efficiency of a monitoring network as a whole, by avoiding having multiple monitors measuring concentrations which are too similar in a single airshed (i.e. above a given threshold of the metric chosen), while aiding in finding a placement of monitors that covers the widest variation in air quality impacts. Further, by comparing the geographic distribution of the model output clusters (i.e., the model airshed maps) against the geographical distribution of monitoring station clusters, we can find areas where
the model disagrees about the airshed distribution, which can aid in directing model development to improve model performance.

Our second example using the NAPS data shows the ease with which this kind of analysis can be performed on very large datasets, without the need to reduce the number of data entries compared through, for example, averaging or selection of a smaller number of stations. The analysis also highlights how similarities in source type across distances of thousands of
kilometers may be identified through clustering of large amounts of observation data.

While we only presented results using the (1-R) metric, the software is currently also able to compute clustering for other metrics, for example the Euclidean distance, $D_{i,j} = \sqrt{\sum\left(x_t^i - x_t^j\right)^2}$, and the product of (1-R) and Euclidean distance metrics (Soares et al., 2018). In principle, it is straightforward to implement other dissimilarity metrics, and because of the code structure and underlying algorithm, we expect no change in the computational performance of the clustering stage. In
addition to the different metrics, the software is currently able to use 7 different linkages (single, complete, average, weighted, Ward's, centroid, and median), that is, methods of computing the new dissimilarity of the combined cluster against the other clusters.

## 5 Conclusions

Given our new HPC-ready parallel implementation of the hierarchical clustering algorithm, it is now possible to perform
clustering analyses on large model-output datasets that was not previously possible. For example, we demonstrated clustering for every grid cell on a full LAM domain of $538 \times 540$ pixels, for a total of 290,520 elements, and a monitoring network analysis across multiple chemical species for 366,427 separate entries  Other analysis methodologies such as



positive matric factorization (PMF) are not capable of analysis with problems of this size due to practical limitations such as the need to invert a very large matrix. To our knowledge this is the first hierarchical clustering analysis tool capable of

handling problems of this size.

## Appendix A

## Construction of the MPI and OpenMP Parallelized Code

We will now describe the overall structure of our clustering program, with special attention to how these parallel programming paradigms were employed.

First, the data is loaded in from long-term storage (usually a hard drive or solid-state drive), and the initial dissimilarity matrix is computed. These steps are performed together because it is possible to save both computation time and memory by computing intermediate components of the dissimilarity metric and then freeing the memory where the raw input data was stored. We apply MPI parallelization here by dividing the dissimilarity matrix up into non-overlapping tiles, and each tile is operated on by a single MPI task. Within each of these independent MPI tasks, there is a for loop over the elements of the

local dissimilarity matrix tile, which are also independent of one another. We therefore parallelize this loop using an OpenMP parallel do directive. Finally at this stage, the tiles of the dissimilarity matrix are output back to storage using MPI parallel I/O which allows the tiles to be stitched back together into a single file. Writing the dissimilarity matrix out at this point allows the code to be restarted for clustering without having to re-compute the initial dissimilarity matrix. This allows for recovery from an unexpected program termination during the second phase, or for splitting up the job across time on

compute clusters with strict wallclock time limits. Because clustering with different linkages starts from the same initial dissimilarity matrix, this output also allows for performing clustering on the same input dataset with different linkages without starting from scratch. It also allows users to run the dissimilarity matrix computation and clustering stages of the program with different parallel setups, which may result in overall lower resource usage.

The second stage of the program uses the dissimilarity matrix calculated in the first stage. Either the data is loaded from

permanent storage or is already in memory, depending on whether the program has been restarted between the two stages, and this is used to populate the pqueues that will be used to store the dissimilarity matrix for the rest of the program. At this stage we also apply MPI parallelization – each MPI task holds a block of pqueues (representing a number of rows of the dissimilarity matrix). To find the minimum value for the whole matrix, each task peeks at all its pqueues (the peek operation returns the lowest value in the pqueue without removing it from the pqueue), finds the pqueue with the minimum of those,

and then each tasks' minima are then compared across tasks to find the task that holds the lowest global value using an MPI_allreduce call. Once every task has the global minimum value and the corresponding $i$ and $j$ (i.e., the indices of the two clusters to be combined), the tasks with the $i^{th}$ and $j^{th}$ clusters must send their pqueues to all the other processes so that they may compute the linkages. At this point the MPI_ROOT task saves the values of $i$, $j$ and $D_{i,j}$ in memory for later output. All tasks then remove both elements referencing clusters $i$ or $j$ from each their pqueues, and then add a new element



representing the new combined cluster to each pqueue. This loop over the local portion of the dissimilarity matrix is performed in parallel using OpenMP parallel do, however the step for inserting the new dissimiliarity values into the pqueues must be wrapped in an OpenMP cricital directive, as multiple threads inserting into the same pqueue at the same time would cause an error. This MPI parallelized loop is repeated until the number of remaining clusters is reduced to one. Finally, the MPI ROOT thread prints out which two clusters were combined and what their dissimilarity value was for all the

clustering steps, which can be used to create the dendrograms and maps presented in the results.

**Code Availability**

The code for the hierarchical clustering program is available online at https://doi.org/10.5281/zenodo.8280445.

**Data Availability**

The data used in the small air quality monitoring network example are publicly available from the Wood Buffalo

Environmental Association (WBEA) at https://wbea.org/data/continuous-monitoring-data/.

The data used in generating the airshed and deposition analysis maps are model results produced at ECCC. The data can be made available on request by contacting Colin.Lee@ec.gc.ca or Paul.Makar@ec.gc.ca.

The NAPS data used in the factor analysis are publicly available at https://donnees-data.ec.gc.ca/data/air/monitor/national-air-pollution-surveillance-naps-program/Data-Donnees/2019/ContinuousData-DonneesContinu/HourlyData-

DonneesHoraires/?lang=en

**Author contributions**

Paul Makar, Joana Soares and Colin Lee designed the experiments. Colin Lee performed the experiments. Colin Lee prepared the manuscript with contributions from Paul Makar and Joana Soares.

**Competing interests**

The authors declare that they have no conflict of interest.

**Acknowledgements**

This work was partially funded under the Oil Sands Monitoring (OSM) Program. It is independent of any position of the OSM Program.





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
