# Peer review of "Clustering analysis of very large measurement and model datasets on high performance computing platforms"

_Geoscientific Model Development, 2023_

## Referee Comment (RC1)

General comments:

The main contribution of this paper is the implementation of hierarchical clustering using OpenMP and MPI parallelization techniques. The problem presented is of size between 10^5 and 10^6 for either measured or modelled air quality variables. Although the proposed algorithm is a clear improvement of the naïve implementation on a single core, there could be further room for improvement using algorithmic techniques or alternative hierarchical clustering methods.

Despite the parallelization on a High-Performance Computing (HPC) infrastructure with many cores, the wall clock time is still of several hours, while an alternative hierarchical clustering method such as accelerated HDBSCAN* would be expected to run in less than a minute for a problem of similar size (or alternatively allows to run problems of magnitude 10^7 to 10^8 in hours if the asymptotic performance is extrapolated from [1705.07321.pdf (arxiv.org)](), Figure 6). As such, a comparison with alternative hierarchical clustering algorithms is needed (see specific comment 1).

The paper does not include many references on clustering large datasets in other fields. Discussion about alternative (hierarchical) clustering algorithms should be included. Three algorithmic techniques for accelerating hierarchical clustering should be discussed (implementation might prove more challenging for the latter two): connectivity constraint (see specific comment 2), efficient data structures (see specific comment 3) and triangular inequality for true distance metrics (see specific comment 4). Other works that have provided implementation of parallelization of clustering algorithms (not necessarily hierarchical) on HPC, multi-thread CPUs or GPUs should be included (see specific comment 5).

I would restructure the paper to reduce the emphasis on "an introduction to hierarchical clustering for non-specialists" and focus more on the practical usage and technical analysis of the hierarchical clustering implementation on OpenMP and MPI (specific comments 6-8, technical comments for L65-73 and L137-195).

Finally, the presentation of the pre-processing of the NAPS dataset on the second example was not done in sufficient details for reproducible results (see specific comment 9).

Specific comments:

1.
   a. Add references and discussion on HDBSCAN*, an alternative hierarchical clustering method with lower algorithmic complexity:
      i. Campello, J.G.B. et al., Density-Based Clustering Based on Hierarchical Density Estimates, LNCS, 2013
      ii. *McInnes, L. et al., Accelerated Hierarchical Density Based Clustering, IEEE International Conference on Data Mining Workshops (ICDMW), 2017 (DOI: [10.1109/ICDMW.2017.12]())*
   b. It is imperative that the authors compare the results against several of the following options to ascertain the value of the proposed implementation:
      i. Scikit-Learn HDBCAN* implementation: [sklearn.cluster.HDBSCAN — scikit-learn 1.3.2 documentation](). A final assignment of noisy points to the closest clusters could be done as a post-processing step to obtain similar results than hierarchical clustering.

ii.   Accelerated HDBSCAN* implementation: GitHub - scikit-learn-contrib/hdbscan: A high performance implementation of HDBSCAN clustering.

iii.  Scikit-learn implementation of hierarchical clustering (as a baseline): sklearn.cluster.AgglomerativeClustering — scikit-learn 1.3.2 documentation

iv.   Scikit-learn implementation of hierarchical clustering, but with grid-cell connectivity constraints (for air quality model data), see A demo of structured Ward hierarchical clustering on an image of coins — scikit-learn 1.3.2 documentation for a usage example.

v.    Scikit-learn implementation of K-means sklearn.cluster.KMeans — scikit-learn 1.3.2 documentation with the same number of clusters as the results presented in the paper for comparison.

c. Considerations can be the following:

i.   For which algorithms it is preferable to use pre-computed pairwise dissimilarity matrix? What is the memory requirement to load this matrix in the RAM? What are the memory constraints of these algorithms that the proposed OpenMP/MPI implementation solves for hierarchical clustering?

ii.  Timing comparison. If the proposed algorithm does not compare favorably to others (it is not expected it will according to algorithmic complexity), maybe it could still be used advantageously for the dissimilarity matrix pre-computation?

iii. How the other results compare to the hierarchical clustering with median linkage and 1-R metric presented in the paper. A quantitative score such as Rand Index could be considered as well as a qualitative comparison. The question is how the clustering results are sensitive to the choice of method (and its optional parameters)? For example, K-means could be very fast, but not very accurate (and losing the flexibility of hierarchical clustering).

iv.  Scaling in function of the number of data points (taking a sub-sample of the dataset) for different number of clusters and data dimension. Presenting the results in log-log plots is the most informative as it allows to easily estimate computational budget for larger datasets.

2. Add discussion on connectivity constraints for clusters:

a. Are all clusters found connected for air quality model data? From the cluster maps (Figures 4 and 5), it appears to be so. It would be worth mentioning if so or analyzing when it does not occur. Can a similar connectivity constraint be found for station data?

b. Employing a connectivity constraint (an implementation equivalent to the connectivity keyword argument in sklearn.cluster.AgglomerativeClustering, see recommendation 1.a.iv) could potentially large speed-up and memory savings. I recommend exploring this possibility for further speed-up of the algorithm. Note however that in this case we would need to be careful with the choice of linkage function (such as Ward's criterion) to avoid "the rich getting richer" phenomenon (getting a few very large clusters and many very small clusters).

3. Add references and discussion on more efficient hierarchical clustering algorithms (see technical comments for L10 for more details on computational efficiency comparison):

a. Improved data structure that speed-up hierarchical clustering (in both theory and practice): *Eppstein,D., Fast Hierarchical Clustering and Other Applications of Dynamic*

> *Closest Pairs, ACM Journal of Experimental Algorithmics, 2000 (https://doi.org/10.1145/351827.351829)*
> b. *Defays, D., An efficient algorithm for complete link method, The Computer Journal, 1977 (https://doi.org/10.1093/comjnl/20.4.364)*

4. If a true distance metric is used such as the Euclidean distance, then the use of the triangle inequality could reduce memory requirements and potentially speed-up the algorithm. The triangular inequality has been used for K-means in 10.1109/ACCESS.2019.2907885, but it has also been explored for hierarchical clustering. Please mention the references and discuss how exploiting the triangular inequality or other data summarization techniques could potentially speed-up computation while reducing memory requirements.
   a. *Zhou J. and Sander, J., Data Bubbles for Non-Vector Data: Speeding-up Hierarchical Clustering in Arbitrary Metric Spaces, Proceedings VLDB Conference, 2003 (https://doi.org/10.1016/B978-012722442-8/50047-1)*
   b. *Kull M., Fast Clustering in Metric Spaces, Master Thesis, 2004 (pdf: content (ut.ee))*

5. Please add references and discussion on other works doing parallelization of clustering algorithms on MPI/OpenMP, multi-thread CPUs or GPUs (also check references therein and paper citing these works):
   a. *Kweldlo W. and Czochanski P.J., A Hybrid MPI/OpenMP Parallelization of K-Means Algorithms Accelerated Using the Triangle Inequality, IEEE Access, 2019 (DOI: 10.1109/ACCESS.2019.2907885)*
   b. *Woodley, A et al., Parallel K-Tree: A multicore, multinode solution to extreme clustering, Future Generation Computer Systems, 2019 (https://doi.org/10.1016/j.future.2018.09.038)*
   c. *Jin C. et al., DiSC: A Distributed Single-Linkage Hierarchical Clustering Algorithm using MapReduce, International Workshop on Data Intensive Computing in the Clouds (DataCloud), November 2013 (pdf: cjinDataCloud13.pdf (northwestern.edu))*

6. Can you expand on why you choose the average linkage function for the examples presented in the paper? Would other linkage functions work as well both in term of computational efficiency and subjective accuracy?

7.
   a. Can you expand on why Pearson's correlation coefficient (1-R) was used as the choice of metric?
   b. This metric will ignore linear transforms (additive and multiplicative shifts in the data), is this a desired feature?
   c. Line 287: Why is the normalization of the species necessary since 1-R is already doing a normalization? Is this step really necessary or I am missing something?

8. The results shown do not take advantage of the hierarchical clustering analysis. Instead, an arbitrary number of clusters is chosen (50 and 100 in the examples). More efficient computation could thus be potentially obtained simply using a highly optimized version of K-means if the number of clusters does not need to be varied. That is, why do we need hierarchical clustering, could other non-hierarchical clustering methods work as well?

9. Not many details were provided on the data pre-processing for stations. Please expand for better reproducibility of the results.

a. How was missing data handled in measurement data? How the algorithm could be used on the COVID-19 year (2020) or on the 209 stations not used in the analysis?
b. Were there any techniques used for quality assurance and quality control of the data, and in particular to remove outliers?
c. Line 286 gives 366,427 data points.
    i. How to arrive at this number? 51 stations x 24 hours x 365 days = 446,760.
    ii. How sensitive the results are to the subsampling of the data? Clearly, it will be costly to perform a sensitivity analysis with the current version of the algorithm on the full dataset and this is why further speed-up of the algorithm would be desirable.
d. Providing the pre-processed subset of NAPS data used in this analysis in open source data repository (and the complete code to automatically generate it) would help to improve the reproducibility of the results.

Technical (line-by-line):

L1: Although it is rather subjective, I would not call a dataset with between 10^5 and 10^6 data points a "very large dataset". For example, by comparing to Table 1 of *Woodley et al. 2019*, we see examples of other works with data sets of size between 10^4 to 10^9 while the work of the referenced paper uses a dataset of size 10^11. To be a bit more precise, I suggest changing the title to "An Implementation of Hierarchical Clustering Analysis on High-Performance Computing Platforms for Large Air Quality Datasets"

L10: "Modern implementations of the algorithm have O(n^2log(n)) computational complexity and memory O(n^2) usage." That statement is not true. For example, even Defays' 1977 CLINK algorithm for complete-linkage hierarchical clustering has a complexity of O(n^2) and O(n^2) memory. Eppstein's 2000 fast hierarchical clustering can achieve O(n^2 log^2 n) time complexity and O(n) space or alternatively O(n^2) time and O(n^2) space. Accelerated HDBCAN* from McInnes 2017 has a time complexity of O(n log n), but it is a slightly different hierarchical clustering approach which excludes some data points as noise/outliers.

L53: Provide citation for hierarchical clustering

L59-60: Provide citation for PMF and K-means

L63: "memory required scales as the number of input data squared" -> only if all pair of distance are pre-computed, alternative implementation can trade-off memory requirements and computational complexity, see comment for line 10.

L65-73: "worse, the computation time scales as the number of input data cubed" -> only for a naïve implementation of the algorithm. Consider removing the example of time and space requirements for the naïve O(n^3) implementation as it should not be considered in practice for larger problems.

L84: "To solve these problems, we turn to parallel computing." -> Although parallel computing on a high-performance computing platform is a sensible solution to squeeze out further performance, it would have been preferable to first seek to optimize the clustering algorithms themselves with appropriate

data structures and constraints or to select a more efficient hierarchical clustering alternative such as HDBSCAN*. Only after this optimization is performed, we should consider parallel computing to push further the performance. I would rephrase.

L113: "HC" -> "High-Performance Computing (HPC)"

L114 and 494: "To our knowledge this is the first time a hierarchical clustering analysis has been performed on such large datasets". -> might be true for air quality data, but certainly not true in general. Although only references to clustering using DBSCAN and K-means are known to have been pushed to the extreme (up to $10^{11}$ data points in Woodley et al. 2019), the accelerated HBDSCAN* hierarchical clustering of McInnes et al. 2017 has been tested for datasets of 200,000 data points (Python implementation running in less than a minute for 50-dimensional data).

L124: "The data could be any sort of 2-dimensional data" -> This statement is confusing as it refers to the fact that the data can be stored in a 2D-array. Usually, the number rows will be the data size (number of samples) whereas the number of columns is the number of dimensions of the data. Please rephrase to clarify that is not the data that is 2-dimensional but that the array in which the data is stored has rows and columns. It could be clarified here that some data can be vectorized to combine several "dimensions" (time, spatial, different variables, etc.).

L137-195: I don't find this example necessary, interesting, or useful. I would replace the example with a more general discussion on hierarchical clustering, the choice of metrics, linkage functions and its different applications in air quality (see specific comments 6-7).

L231: "operated by Shared Services Canada" -> that might not be clear to what is refers to a general audience, might want to add "the department responsible for providing computing infrastructure for the Government of Canada".

L298: "which would be prohibitive using (…) K-means" -> Efficient implementations of K-means scale generally better than hierarchical clustering, so this is not a fair comparison. For example, see Woodley et al. 2019, Table 1 and McInnes et al., 2017, Figure 6.

L304: Isn't the wall clock limit 6 hours on the current Shared Services Canada infrastructure?

L451: "The authors would like to emphasize that there may be many considerations required to obtain the best performance on any given high-performance computing cluster." -> could you enumerate a few of them?

---

## Author Comment (AC2)

**Response to Reviewer's Comments, "Clustering analysis of very large measurement and model datasets on high performance computing platforms"**

We thank the reviewers for their detailed feedback. The main message from both reviewers appears to be that our work needs to have a greater focus on the data analysis aspects, and its context within the broader literature of clustering and other data analysis literature. We appreciate the additional references and can incorporate a broader discussion of the techniques the reviewers mentioned, and their pros and cons relative to the approach we have taken, in the Introduction of the material. We also note that our inclusion of "introductory" material was in our submitted draft with the intent of making the work more accessible to those readers of GMD who may lack background in hierarchical clustering – rather than remove that information altogether, we can move it to the Supporting Information, referencing it from the main document, and thus freeing up more space to discuss related methodologies in more detail. We still believe that this implementation of hierarchical clustering is a novel and useful addition to the literature, and that the number of other publications and software tools designed to speed up or otherwise improve clustering speaks to the importance of the subject to the community.

In order to improve the presentation beyond our submitted draft, we have taken initial steps towards, and would like to take the opportunity to work on, further exploration of some of the algorithms presented in the reviewers' comments. We would like to submit a revised draft based on this exploration. This additional work should be able to be completed within a month of the receipt of this response. We have done some preliminary tests using the HDBSCAN* algorithm, which was mentioned by both reviewers. Our results indicate that while the algorithm provides very quick results for low-rank datasets, as evidenced by the 2D cases presented in [1705.07321.pdf (arxiv.org)](), we have found that the algorithm scales slowly with the dimensionality of the data. The HDBSCAN* algorithm operates efficiently for our 7-dimensional test case (the NAPS observation dataset example supplied in our original draft), completing clustering in just under 18 minutes, but this speedup diminishes quickly as the dimensionality of the problem increases. This is particularly pertinent to our GEM-MACH simulation output case study – based on extrapolated timing curves from our system, a problem with a dimensionality of 10968 (24-timesteps per day x 15 months), running HDBSCAN* on our full model domain of 290,520 points, would take approximately 8 days of wallclock time on our compute cluster, compared to 13 hours with our hierarchical clustering algorithm. We have attached a figure of these timing curves,

with the full model timeseries being represented by the D=10680 curve. We also achieved similar timing results for the python fastcluster library, which we also project to require roughly 8 days to cluster our full simulation dataset.

[Figure]

*Figure 1 Timing curves for HDBSCAN\* on subsets of our GEM-MACH model output set in time (D) and space (N).*

We note that many of the algorithms and implementations presented by the reviewers are either limited to specific metrics or linkages, require simplifying assumptions, or provide less general results than are produced by our implementation of hierarchical clustering. One of the main aims of our work was to reduce processing time via more efficient construction of a general algorithm, as opposed to reducing processing time via algorithm simplifications.  For example, while the HDBSCAN\* produces a list of clustered points, plus a list of outliers which could not be clustered by the algorithm, the selection of which points are to be considered outliers is a function of the input parameters provided by the user when running the algorithm.  In 2 or even 3 dimensions it is fairly easy to verify that the clusters are indeed clusters and the outliers are indeed outliers – however, it is much harder to visualize 7 or 10680

dimensions to verify this visually, and to ensure that important information has not been lost in the clustering process. An alternative approach which might achieve the same information from HDBSCAN* would be conducting a set of iterative clusterings until a suitable solution is found, although this may somewhat decrease the computation benefit it provides at lower dimensionalities. With hierarchical clustering, the entire dendrogram is generated once, and outliers may be identified via a post-processing step by finding an appropriate dissimilarity level at which to cut the dendrogram. In the case of our 10680-dimensional air quality simulation output dataset, this allows for visualising the complex high-dimensional data as an interactive map which can be animated by changing the value of the dissimilarity metric threshold. An advantage of the approach we have taken aside from its computational efficiency is that no prior knowledge of an appropriate level of clustering or outliers is required – it is the generation of the entire dendrogram in a single set of calculations allows the user to gain that knowledge.

Nevertheless, we agree with the reviewers that we should present our results in the context of other clustering algorithms, to highlight what new analyses our implementation makes possible, while also discussing the ways in which the results produced by this algorithm differ from other clustering approaches.

We have begun numerical investigation and a revised manuscript to address these issues. As we mentioned above it will take approximately a month to perform the additional analysis required to fully address the issues raised by the reviewers.

Responses to specific comments are presented in blue below the original reviewer comment.

We thank the editor and reviewers for their time and efforts and hope that our responses herein will make the case for the submission of a revised paper, better positioning this work as novel yet within the context of other methodologies present in the literature.

**Response to reviewer 1 (responses in italics)**

General comments:

The main contribution of this paper is the implementation of hierarchical clustering using OpenMP and MPI parallelization techniques. The problem presented is of size between 10^5 and 10^6 for either measured or modelled air quality variables. Although the proposed algorithm is a clear improvement of the naïve implementation on a single core, there could be further room for improvement using algorithmic techniques or alternative hierarchical clustering methods.

Despite the parallelization on a High-Performance Computing (HPC) infrastructure with many cores, the wall clock time is still of several hours, while an alternative hierarchical clustering method such as accelerated HDBSCAN* would be expected to run in less than a minute for a problem of similar size (or alternatively allows to run problems of magnitude 10^7 to 10^8 in hours if the asymptotic performance is extrapolated from [1705.07321.pdf (arxiv.org)](https://arxiv.org), Figure 6). As such, a comparison with alternative hierarchical clustering algorithms is needed (see specific comment 1).

*We thank the reviewers for pointing us to HDBSCAN*.  We were able to apply the Scikit implementation to our test cases:  our initial analysis suggests that the algorithm performs well for problems with lower levels of dimensionality (e.g. the 7-dimensional NAPS data case where the 7 dimensions are the different pollutants), but relatively poorly compared to our algorithm for large-scale problems such as our air-quality model analysis, where the dimensionality is much higher (10968).  The latter problem can be easily reconfigured for different levels of dimensionality; we propose to investigate this aspect and compare the algorithms directly as a function of dimensionality in the revised manuscript. We also note that this is a different type of clustering, density clustering, and will therefore produce different results than our hierarchical clustering algorithm. We propose to add an analysis of the differences in the results on our datasets to the revised manuscript. HDBSCAN* also produces what is effectively a single cut of the dendrogram, while hierarchical clustering produces the entire dendrogram, allowing for a single run of hierarchical clustering to provide more information for analysis after the fact.*

The paper does not include many references on clustering large datasets in other fields. Discussion about alternative (hierarchical) clustering algorithms should be included. Three algorithmic techniques for accelerating hierarchical clustering should be discussed (implementation might prove more challenging for the latter two): connectivity constraint (see specific comment 2), efficient data structures (see specific comment 3) and triangular inequality for true distance metrics (see specific comment 4). Other works that have provided implementation of parallelization of clustering algorithms (not necessarily hierarchical) on HPC, multi-thread CPUs or GPUs should be included (see specific comment 5).

*We will include alternative clustering methods (e.g. HDBSCAN* density clustering and K-means, as well as comparing CLINK and SLINK).  However, we also note that some implementation approaches may require assumptions, simplifications, or limitations compared to the approach taken in our submitted initial draft, and may thus provide modified results.  We propose to discuss alternative approaches and their benefits and disadvantages in the revised work.*

I would restructure the paper to reduce the emphasis on "an introduction to hierarchical clustering for non-specialists" and focus more on the practical usage and technical analysis of the hierarchical

clustering implementation on OpenMP and MPI (specific comments 6-8, technical comments for L65-73 and L137-195).

*In making our initial submission, we felt that many of the readers of GMD would be unfamiliar with the overall use of clustering algorithms (particularly for those more familiar with air-quality chemical transport models such as the one used for our high dimensionality example). For the benefit of those readers, we propose moving the introductory material to the Supplemental Information portion of the paper, reducing it in the main body to a reference to the supplemental for those desiring introductory material. We will replace this in the main body by more information on the practical usage and technical analysis as recommended by Reviewer 1.*

Finally, the presentation of the pre-processing of the NAPS dataset on the second example was not done in sufficient details for reproducible results (see specific comment 9).

*A fair point: we propose including the pre-processing script we used on the NAPS data in our zenodo response. The issues raised by the reviewer are easy to address with additional text within the paper. In choosing the stations for analysis we were attempting to provide stations which had concurrent measurements of the same species for the largest number of measurement hours – which resulted in 51 stations being selected. Not all stations in the NAPS network provide the same information at all stations, nor provide them at comparable times. The number of stations was thus reduced – however, we agree that this information should be clarified in the manuscript. We note that the intent here was to provide a useful example of the clustering algorithm for factor analysis using a publicly available dataset.*

*Brief responses:*

    a.  How was missing data handled in measurement data? How the algorithm could be used on the COVID-19 year (2020) or on the 209 stations not used in the analysis?
        *Missing data was simply excluded. There is nothing precluding the use of the algorithm on 2020, but the number of hours observed for that year was much smaller than for 2019. Because the example we presented was factor analysis, what we were looking for was concurrent measurements of several pollutants. By decreasing the number of pollutants, we could include more measurement hours, which would presumably increase the number of stations we could use in the analysis. The main aim was to provide a problem of a reasonably large size, given those constraints, as an example use of our methodology.*

    b.  Were there any techniques used for quality assurance and quality control of the data, and in particular to remove outliers?
        *We note that there are different meanings of the word "outlier" depending on the field (data analysis versus air-quality observations). In the latter, an outlier refers to data which in some way is suspect, the result of measurement error. Under that latter interpretation, we note that the NAPS data undergoes quality assurance and control as part of the generation of the database; the numbers released and used within our test case are QA/QC'd data. We can provide references for NAPS' QA/QC procedures (for example https://ccme.ca/en/res/ambientairmonitoringandqa-qcguidelines_ensecure.pdf ). These include data flags identifying outliers (over-range values, see section 12.5.4,*

*page 64 of the above reference). If the former interpretation (e.g. "outliers" in the context of identification of more unique but valid datapoints which are different from the others in the data), no a priori specification by the user on criteria for determining outliers is required.*

    c. Line 286 gives 366,427 data points.

        i. How to arrive at this number? 51 stations x 24 hours x 365 days = 446,760.

*Not all stations provide data every 24 hours; for example automatic instrument calibration events, pre-determined QA/QC may remove some data, and simple instrumentation failure may contribute to the lack of records. As noted above, our choice of number of stations was determined by the desire to have stations with a common set of measured species at as many times as possible in the measurement record.*

        ii. How sensitive the results are to the subsampling of the data? Clearly, it will be costly to perform a sensitivity analysis with the current version of the algorithm on the full dataset and this is why further speed-up of the algorithm would be desirable.

*The "subsampling of the data" was not intended as means of reducing the information quantity analysed – rather, our intent was merely to show how the algorithm can be used, in this case for factor analysis. We chose to do this with stations and times that reported a suite of 7 pollutants concurrently, but this could be easily changed. The reviewers' inference that the "full" dataset would be difficult to analyse is incorrect – we can repeat the analysis with the full dataset fairly easily (and at low cost). We thought the given subset was sufficient to demonstrate a possible use (and processing time) for our algorithm – there was no intent of reducing the size of the problem to make it more tractable. Note that our other test example makes use of a much larger dataset.*

    d. Providing the pre-processed subset of NAPS data used in this analysis in open source data repository (and the complete code to automatically generate it) would help to improve the reproducibility of the results.

*The NAPS data we used was from a public, open-data repository. We will include the scripts we used to pre-process the data as part of the revised version.*

Specific comments:

    1.

        a. Add references and discussion on HDBSCAN*, an alternative hierarchical clustering method with lower algorithmic complexity:

            i. Campello, J.G.B. et al., Density-Based Clustering Based on Hierarchical Density Estimates, LNCS, 2013 ii. *McInnes, L. et al., Accelerated Hierarchical Density Based Clustering, IEEE*

            *International Conference on Data Mining Workshops (ICDMW), 2017 (**DOI:** 10.1109/ICDMW.2017.12)*

*Easy to do – see also our comments above regarding our initial tests with HDBSCAN*.*

b. It is imperative that the authors compare the results against several of the following options to ascertain the value of the proposed implementation:

i. Scikit-Learn HDBCAN* implementation: sklearn.cluster.HDBSCAN — scikit-learn 1.3.2 documentation. A final assignment of noisy points to the closest clusters could be done as a post-processing step to obtain similar results than hierarchical clustering.

ii. Accelerated HDBSCAN* implementation: GitHub - scikit-learn-contrib/hdbscan: A high performance implementation of HDBSCAN clustering.

iii. Scikit-learn implementation of hierarchical clustering (as a baseline): sklearn.cluster.AgglomerativeClustering — scikit-learn 1.3.2 documentation

iv. Scikit-learn implementation of hierarchical clustering, but with grid-cell connectivity constraints (for air quality model data), see A demo of structured Ward hierarchical clustering on an image of coins — scikit-learn 1.3.2 documentation for a usage example.

v. Scikit-learn implementation of K-means sklearn.cluster.KMeans — scikit-learn 1.3.2 documentation with the same number of clusters as the results presented in the paper for comparison.

c. Considerations can be the following:

i. For which algorithms it is preferable to use pre-computed pairwise dissimilarity matrix? What is the memory requirement to load this matrix in the RAM? What are the memory constraints of these algorithms that the proposed OpenMP/MPI implementation solves for hierarchical clustering?

ii. Timing comparison. If the proposed algorithm does not compare favorably to others (it is not expected it will according to algorithmic complexity), maybe it could still be used advantageously for the dissimilarity matrix pre-computation? iii. How the other results compare to the hierarchical clustering with median linkage and 1-R metric presented in the paper. A quantitative score such as Rand Index could be considered as well as a qualitative comparison. The question is how the clustering results are sensitive to the choice of method (and its optional parameters)? For example, K-means could be very fast, but not very accurate (and losing the flexibility of hierarchical clustering).

iii. Scaling in function of the number of data points (taking a sub-sample of the dataset) for different number of clusters and data dimension. Presenting the results in log-log plots is the most informative as it allows to easily estimate computational budget for larger datasets.

*As noted above, we have already carried out some initial analysis making use of HDBSCAN* (both the stock scikit learn version and the contrib version). We've found that the relative performance of that algorithm versus that ours depends on the dimensionality of the problem posed.  As noted by the reviewer, some of the approaches to obtain faster cpu-times may come at the cost of accuracy – for this reason, our choice of algorithm implementation with simple averaging was intended to provide high accuracy without the use of simplifications.  We are also specifically designing for users with access to high performance computing clusters with a large number of processors – the need for simplifications*

*may be less necessary over time. Nevertheless, we can carry out some of the tests suggested by the reviewer.*

- *Compare our results against K-means results for the largest possible subset of our data*
- *Compare our results against HDBSCAN\* result for the largest possible subset of our data*
- *Compare our timing and memory usage results against CLINK and SLINK*

*Also noted above and in the next comment, 2.a, is that outlier detection is more of a post-analysis step with hierarchical clustering. Our method has the advantage of providing the user with all levels of the dendrogram, which in turn allows the user to determine the manner in which outliers occur, and the level of the metric at which they occur. This is certainly worth deeper discussion in the revised manuscript.*

2. Add discussion on connectivity constraints for clusters:
   a. Are all clusters found connected for air quality model data? From the cluster maps (Figures 4 and 5), it appears to be so. It would be worth mentioning if so or analyzing when it does not occur. Can a similar connectivity constraint be found for station data?

*Because this is hierarchical clustering, all data are clustered in some sense. However, we could consider clusters of size 1 to be "unconnected" clusters for a given threshold of the dissimilarity metric. This is a major advantage hierarchical clustering which we intend to emphasize in the revised manuscript – after running the clustering a single time, you have all the possible groupings available by simply changing the dissimilarity threshold, ie, the level at which you cut the dendrogram.*

   b. Employing a connectivity constraint (an implementation equivalent to the connectivity keyword argument in sklearn.cluster.AgglomerativeClustering, see recommendation 1.a.iv) could potentially large speed-up and memory savings. I recommend exploring this possibility for further speed-up of the algorithm. Note however that in this case we would need to be careful with the choice of linkage function (such as Ward's criterion) to avoid "the rich getting richer" phenomenon (getting a few very large clusters and many very small clusters).

*This is a fair point, and warrants some investigation. Our concerns with implementing this at this time are how it might impact the parallelisability of the code, and how the connectivity constraint would impact the results with high-dimensional data. It's easy to understand (and visualize) how the connectivity constraint works in 3 dimensions with a Euclidean distance metric, but less obvious how a constraint like this would affect clustering in 11,000 dimensions and with other metrics such as correlation.*

3. Add references and discussion on more efficient hierarchical clustering algorithms (see technical comments for L10 for more details on computational efficiency comparison):
   a. Improved data structure that speed-up hierarchical clustering (in both theory and practice): *Eppstein,D., Fast Hierarchical Clustering and Other Applications of Dynamic Closest Pairs, ACM Journal of Experimental Algorithmics, 2000 (https://doi.org/10.1145/351827.351829)*
   b. *Defays, D., An efficient algorithm for complete link method, The Computer Journal, 1977 (https://doi.org/10.1093/comjnl/20.4.364)*

*Certainly – these can be added to the introductory material as well as some discussion there of the pros and cons of the different approaches.*

4.  If a true distance metric is used such as the Euclidean distance, then the use of the triangle inequality could reduce memory requirements and potentially speed-up the algorithm. The triangular inequality has been used for K-means in 10.1109/ACCESS.2019.2907885, but it has also been explored for hierarchical clustering. Please mention the references and discuss how exploiting the triangular inequality or other data summarization techniques could potentially speed-up computation while reducing memory requirements.

    a.  *Zhou J. and Sander, J., Data Bubbles for Non-Vector Data: Speeding-up Hierarchical Clustering in Arbitrary Metric Spaces, Proceedings VLDB Conference, 2003 (https://doi.org/10.1016/B978-012722442-8/50047-1)*

    b.  *Kull M., Fast Clustering in Metric Spaces, Master Thesis, 2004 (pdf: content (ut.ee))*

*Certainly. We should also mention the potential limitations of these methodologies. For example, the Euclidean distance is one possible metric, though others (such as 1-R as used in our analysis) are possible, as are combined metrics. Our code can be used with multiple distance metrics and can easily be extended to include others.*

5.  Please add references and discussion on other works doing parallelization of clustering algorithms on MPI/OpenMP, multi-thread CPUs or GPUs (also check references therein and paper citing these works):

    a.  *Kweldlo W. and Czochanski P.J., A Hybrid MPI/OpenMP Parallelization of K-Means Algorithms Accelerated Using the Triangle Inequality, IEEE Access, 2019 (**DOI:** 10.1109/ACCESS.2019.2907885)*

    b.  *Woodley, A et al., Parallel K-Tree: A multicore, multinode solution to extreme clustering, Future Generation Computer Systems, 2019 (https://doi.org/10.1016/j.future.2018.09.038)*

    c.  *Jin C. et al., DiSC: A Distributed Single-Linkage Hierarchical Clustering Algorithm using MapReduce, International Workshop on Data Intensive Computing in the Clouds (DataCloud), November 2013 (pdf: cjinDataCloud13.pdf (northwestern.edu))*

*Certainly. Again, these will be added to the Introduction.*

6.  Can you expand on why you choose the average linkage function for the examples presented in the paper? Would other linkage functions work as well both in term of computational efficiency and subjective accuracy? *The average is one of the most intuitive linkage functions, but our implementation of the algorithm includes all 7 common linkage functions, and in our testing before submission of the original manuscript, choice of linkage did not impact efficiency.*

7.

    a.  Can you expand on why Pearson's correlation coefficient (1-R) was used as the choice of metric?

    *As the focus of the paper was demonstration of the algorithm on air quality datasets, we used the (1-R) metric which was shown to provide meaningful results for airshed mapping in our referenced previous work (which also made cases for Euclidean distance and a product of (1-R) and Euclidean distance). There is no reason any other metric should change our timing results, as the creation of the dissimilarity matrix does not contain the lion's share of the processing time, although they will obviously change the clustering results. This may be worth exploring if there is space in the revised manuscript.*

b. This metric will ignore linear transforms (additive and multiplicative shifts in the data), is this a desired feature? *True – in the context of an air-quality analysis, the issue the reviewer identifies can manifest as a plume measured at two different distances downwind by a pair of monitoring stations having a between-station correlation of unity, but the magnitudes of concentrations may be very different, the downwind station having very low concentrations compared to the station closer to the source. However, by the same token, the 1-R metric may thus identify such locations that are affected by the same sources, regardless of proximity. The choice of a metric may thus depend on the desired information to be gained. The algorithm is easily modified for other metrics.*

c. Line 287: Why is the normalization of the species necessary since 1-R is already doing a normalization? Is this step really necessary or I am missing something? *The test case here (factor analysis on NAPS data) looks across different chemical species, some of which may be observed in very different units and hence the numerical values may vary by orders of magnitude – this is illustrated by the fact that we had to multiply CO and SO$_2$ by 100 and 10, respectively, in order to plot them on the same y axis in figure 10. Normalization in this case attempts to prevent species with high real number values (in one set of units) from having outsized effects on the results of the suite of species.*

8. The results shown do not take advantage of the hierarchical clustering analysis. Instead, an arbitrary number of clusters is chosen (50 and 100 in the examples). More efficient computation could thus be potentially obtained simply using a highly optimized version of K-means if the number of clusters does not need to be varied. That is, why do we need hierarchical clustering, could other non-hierarchical clustering methods work as well?

*We have found in practical applications (c.f. Soares et al paper referenced in our original submission) that one of the questions we have to answer is "What is the appropriate number of clusters?" or "At what levels in the metric are stations clustering?" Knowing that a large ground of stations cluster at an R level of 0.95 for example tells us that with respect to correlation, some of those stations may be redundant, resulting in considerable savings and/or a better deployment of station locations. Hierarchical clustering provides a full analysis of the data – and that information is useful for analysing AQ data – making no assumptions on the methodology allows features that might unexpected to be identified.*

9. Not many details were provided on the data pre-processing for stations. Please expand for better reproducibility of the results.

a. How was missing data handled in measurement data? How the algorithm could be used on the COVID-19 year (2020) or on the 209 stations not used in the analysis?

b. Were there any techniques used for quality assurance and quality control of the data, and in particular to remove outliers?

c. Line 286 gives 366,427 data points.

   i. How to arrive at this number? 51 stations x 24 hours x 365 days = 446,760.

   ii. How sensitive the results are to the subsampling of the data? Clearly, it will be costly to perform a sensitivity analysis with the current version of the algorithm on the full dataset and this is why further speed-up of the algorithm would be desirable.

d.  Providing the pre-processed subset of NAPS data used in this analysis in open source data repository (and the complete code to automatically generate it) would help to improve the reproducibility of the results.

*See above responses.*

Technical (line-by-line):

L1: Although it is rather subjective, I would not call a dataset with between 10^5 and 10^6 data points a "very large dataset". For example, by comparing to Table 1 of *Woodley et al. 2019*, we see examples of other works with data sets of size between 10^4 to 10^9 while the work of the referenced paper uses a

dataset of size 10^11. To be a bit more precise, I suggest changing the title to "An Implementation of Hierarchical Clustering Analysis on High-Performance Computing Platforms for Large Air Quality Datasets"

*We will consider this carefully and examine the literature with an eye to the sizes of datasets being clustered before revising the manuscript.*

L10: "Modern implementations of the algorithm have $O(n^2\log(n))$ computational complexity and memory $O(n^2)$ usage." That statement is not true. For example, even Defays' 1977 CLINK algorithm for complete-linkage hierarchical clustering has a complexity of $O(n^2)$ and $O(n^2)$ memory. Eppstein's 2000 fast hierarchical clustering can achieve $O(n^2 \log^2 n)$ time complexity and $O(n)$ space or alternatively $O(n^2)$ time and $O(n^2)$ space. Accelerated HDBCAN* from McInnes 2017 has a time complexity of $O(n \log n)$, but it is a slightly different hierarchical clustering approach which excludes some data points as noise/outliers.

*Given our preliminary findings described above, the complexity also appears to involve the dimensionality of the data (this is noted in some but not all of the references describing the original algorithms). We will have to consider this carefully and present a full accounting of how these input sizes impact algorithm performance.*

L53: Provide citation for hierarchical clustering

*We will add this.*

L59-60: Provide citation for PMF and K-means

*We will add these.*

L63: "memory required scales as the number of input data squared" -> only if all pair of distance are precomputed, alternative implementation can trade-off memory requirements and computational complexity, see comment for line 10.

*Fair point. See above, we will improve our discussion of the complexity.*

L65-73: "worse, the computation time scales as the number of input data cubed" -> only for a naïve implementation of the algorithm. Consider removing the example of time and space requirements for the naïve $O(n^3)$ implementation as it should not be considered in practice for larger problems.

*We do note that "Modern implementations of the algorithm have $O(n^2 \log(n))$ computational complexity and $O(n^2)$ memory usage" but we will need to strengthen this discussion, as there are some algorithms which appear to have $O(n \log(n))$ or $O(n^2)$ time complexity and $O(n^2)$ or $O(n)$ memory usage. It is possible that some of the discrepancy here comes from worst case vs average case, but we will need to examine this discussion closely.*

L84: "To solve these problems, we turn to parallel computing." -> Although parallel computing on a high performance computing platform is a sensible solution to squeeze out further performance, it would have been preferable to first seek to optimize the clustering algorithms themselves with appropriate data structures and constraints or to select a more efficient hierarchical clustering alternative such as HDBSCAN*. Only after this optimization is performed, we should consider parallel computing to push further the performance. I would rephrase.

*We agree that we can make a better case for our solution by providing comparisons against some of the other methods to demonstrate which trade-offs need to be made in terms of assumptions, simplifications, heuristics and the output results.*

L113: "HC" -> "High-Performance Computing (HPC)"

L114 and 494: "To our knowledge this is the first time a hierarchical clustering analysis has been performed on such large datasets". -> might be true for air quality data, but certainly not true in general. Although only references to clustering using DBSCAN and K-means are known to have been pushed to the extreme (up to 10^11 data points in Woodley et al. 2019), the accelerated HBDSCAN* hierarchical clustering of McInnes et al. 2017 has been tested for datasets of 200,000 data points (Python implementation running in less than a minute for 50-dimensional data).

*Fair enough, we will temper our statement and add appropriate qualifiers.*

L124: "The data could be any sort of 2-dimensional data" -> This statement is confusing as it refers to the fact that the data can be stored in a 2D-array. Usually, the number rows will be the data size (number of samples) whereas the number of columns is the number of dimensions of the data. Please rephrase to clarify that is not the data that is 2-dimensional but that the array in which the data is stored has rows and columns. It could be clarified here that some data can be vectorized to combine several "dimensions" (time, spatial, different variables, etc.).

*This language can be tightened to be more precise.*

L137-195: I don't find this example necessary, interesting, or useful. I would replace the example with a more general discussion on hierarchical clustering, the choice of metrics, linkage functions and its different applications in air quality (see specific comments 6-7).

*See above.*

L231: "operated by Shared Services Canada" -> that might not be clear to what is refers to a general audience, might want to add "the department responsible for providing computing infrastructure for the Government of Canada".

*Thank you we will provide this clarification.*

L298: "which would be prohibitive using (…) K-means" -> Efficient implementations of K-means scale generally better than hierarchical clustering, so this is not a fair comparison. For example, see Woodley et al. 2019, Table 1 and McInnes et al., 2017, Figure 6.

*We will examine this in the revised manuscript.*

L304: Isn't the wall clock limit 6 hours on the current Shared Services Canada infrastructure?

*It depends on the system. Initially we were performing our analysis on a system with a 4.5 hour time limit, but we have also done it on a system with a 6 hour time limit.*

L451: "The authors would like to emphasize that there may be many considerations required to obtain the best performance on any given high-performance computing cluster." -> could you enumerate a few of them?

*Yes, we will provide this in a revised manuscript.*

Response to reviewer 2

Dear Author(s),

Firstly, I would like to commend your efforts in developing a hierarchical clustering method based on the integration of MPI and OpenMP. The application of your method to air-quality model analysis, and its efficacy on large-scale data in a distributed memory architecture cluster, is indeed noteworthy.

However, before this paper can be considered outstanding, there are several areas that could be improved upon. I would like to highlight these and provide some suggestions:

1、 The paper could benefit from a more comprehensive review of related work on the parallelization of hierarchical clustering and other clustering methods. There is existing literature where scholars have made efforts to parallelize hierarchical clustering on distributed clusters, such as "A scalable algorithm for single-linkage hierarchical clustering on distributed-memory architectures," "A novel parallelization approach for hierarchical clustering," and "A hierarchical clustering algorithm for MIMD architecture." The current version of the manuscript seems to lack a thorough survey of these related studies.

*Thank you for these references, we will include them in the introduction of the revised manuscript. As we mentioned above, these references seem to indicate a need within the community for high-performance implementations such as we are presenting here.*

2、 In lines 80 to 86, the manuscript discusses the shift towards parallel computing to address the issues of high time and space complexity associated with hierarchical clustering of ultra-large scale data. However, I would suggest considering existing high-performance hierarchical clustering algorithms that reduce complexity at the algorithmic level before resorting to increased computational resources, which could be more effective. Improvements could be further amplified when combined with the additional computational resources. Several high-performance algorithms have already been optimized for time and space complexities in the field of hierarchical clustering. Here are a few notable methods:

a. SLINK (Single-Linkage Clustering): Optimized for single-linkage hierarchical clustering, SLINK algorithm reduces the time complexity to $O(n^2)$ and space complexity to $O(n)$.

b. CLINK (Complete-Linkage Clustering): Similar to SLINK, optimized for complete-linkage clustering with a time complexity of $O(n^2)$, although it may be slower with

different data types.

c. Fastcluster: An efficient hierarchical clustering library for larger datasets, providing efficient implementations for different linkage strategies such as single, complete, and average linkage.

d. BIRCH (Balanced Iterative Reducing and Clustering using Hierarchies): Designed for large datasets, BIRCH initially compresses data using a CF Tree before applying hierarchical clustering.

e. HDBSCAN (Hierarchical Density-Based Spatial Clustering of Applications with Noise): A density-based clustering algorithm extending DBSCAN, converting the concept of core distance into hierarchical clustering to identify clusters of varying density.

f. Efficient Agglomerative Clustering using a Heap (EAC): EAC uses a heap data structure to optimize the process of finding the nearest cluster pair, making the merging steps more efficient.

g. Divisive Analysis Clustering (DIANA): A divisive clustering algorithm that starts with a cluster of all samples and recursively splits it into smaller clusters, which may be more efficient for large datasets compared to agglomerative methods.

*As we mentioned above, each of these approaches has limitations, drawbacks, underlying assumptions about the input data, or heuristics, or is simply a different type of clustering, and therefore will not produce the same general-purpose cluster-once analyse-after result produced by hierarchical clustering. We certainly agree that comparing some of these methods to our method will help make the case for our new implementation of hierarchical clustering which implements all 7 common linkages and can easily accommodate different dissimilarity metrics.*

3、The manuscript would also benefit from an expanded discussion on the necessity of using the hierarchical clustering method. Readers not specialized in the field may wonder why one must use a hierarchical clustering method, which is relatively more complex, over other faster clustering methods such as K-means. A balance between effectiveness and efficiency could be better articulated.

*As we have already discussed, we agree that this will help our narrative and thank you for the suggestion.*

4、Additional experimentation would strengthen the paper's contribution. Comparisons of the proposed method with previous parallel hierarchical clustering

algorithms, traditional high-performance hierarchical clustering methods like HDBSCAN, and simpler clustering methods such as K-means, would be highly illustrative. Such comparisons would highlight the necessity, advantages, and advancements of your work.

*See above.*

Regrettably, considering the aforementioned points, I believe the article is not yet suitable for acceptance in its current form.

Best regards